# Optimized Selection of Water Resource Allocation Schemes Based on Improved Connection Entropy in Beijing’s Southern Plain

**DOI:** 10.3390/e24070920

**Published:** 2022-07-01

**Authors:** Chen Li, Baohui Men, Shiyang Yin

**Affiliations:** College of Water Resources and Hydropower Engineering, North China Electric Power University, Beijing 102206, China; lichen5969@sina.com (C.L.); menbh@ncepu.edu.cn (B.M.)

**Keywords:** connection entropy, water resource allocation scheme, water quality evaluation, groundwater, Beijing

## Abstract

Increased urbanization has caused problems such as increasing water consumption and the continuous deterioration of the groundwater environment. It is necessary to consider the groundwater quality in the water resource optimization system and increase the rate of reclaimed water development to reduce the amount of groundwater exploitation and achieve sustainable development of water resources. This study used the Daxing District, a region of Beijing’s southern plain, as an example to evaluate water quality by analyzing water quality data of surface and groundwater from 2012 to 2016 and actual water-use schemes from 2006 to 2016. Three groundwater extraction modes were set up based on NO_3_–N concentrations, water resources were optimized under three extraction modes, and water resource optimization schemes were determined based on the improved connection entropy. The results show that (1) the surface water quality was poor, and the proportion of V_4_ type water in the indexes of NH_3_–N and chemical oxygen demand (COD) was the largest. The surface water can only be used for agricultural irrigation. The pollution sources contributing most to NH_3_–N and COD were domestic and agricultural pollution sources. (2) The groundwater quality was good. The NO_3_–N index was primarily type I–III water, accounting for 95.20% of the total samples. Severe NH_3_–N pollution areas were mainly in the northern region, and most regional groundwater can be used for various purposes. (3) Taking 2016 as an example, three groundwater exploitation modes were set to optimize water resource allocation, and the results showed that the rate of groundwater development and NO_3_–N pollution decreased significantly after optimization. (4) Connection entropy is an evaluation method that combines connection numbers and entropy, including identify, difference, and opposition entropy. As connection entropy being a kind of complete entropy, which can reflect the difference of the system in different states, based on the improved connection entropy, the connection entropies of optimal water resource allocation and actual water-use schemes were calculated. The connection entropy of groundwater exploitation mode 3 was less than that of groundwater exploitation modes 1 and 2 and actual water-use schemes from 2006 to 2016. Therefore, exploitation mode 3’s water resource optimization scheme was recommended. In the paper, satisfactory results have been obtained. As a kind of complete entropy, connection entropy has great research value in dealing with complex hydrological problems. This study’s research methods and outcomes can provide methodological and theoretical lessons for water management in freshwater-deficient areas.

## 1. Introduction

China’s increasing urbanization and rapid economic and social development in the past 20 years have brought about many environmental problems [1,2]. Beijing is in the north of North China. Groundwater has been the city’s primary source of water supply, leading to a continuous decline in groundwater level, land subsidence, and deteriorating groundwater quality [3,4,5,6]. Reclaimed water is the second stable water source in the city. Although the water quality is not as good as conventional water sources, it can meet industrial, agricultural, ecological, and municipal miscellaneous water standards. It is a necessity to replace fresh water and reduce groundwater exploitation [7,8].

According to the Beijing Water Statistics Yearbook from 2003 to 2020, the proportion of reclaimed water supply in Beijing increased from 5.73% to 29.56%, indicating that reclaimed water is vital in the existing water supply pattern. Although reclaimed water is increasingly important, more than 90% of reclaimed water is used for river water supplement. Because of the insufficient coverage of reclaimed water pipelines or psychological effects [9,10], reclaimed water is not used for other purposes, resulting in a waste of water resources. Due to the severe water environment problems caused by groundwater overexploitation, some problems exist in the water resource allocation schemes, which must be optimized. Therefore, how to further tap the water supply capacity of different water sources according to water supplies of different qualities [11,12] to improve the groundwater environment and realize the sustainable use of water resources must be further studied.

Currently, research on optimal water resource allocation is divided into three types. The first type is to determine the water supply based on water demand. Due to this single consideration, it is harmful to improve water resource use efficiency [13]. The second type simultaneously considers the water demand and economy of different water-use types [14]. The third type considers the ecological benefits of water resources when considering the water demand and economy of various water uses [15]. With further research on optimal water resource allocation, many scholars have considered more details such as the fairness of the water supply [16], the risk of water supply shortage [17,18], and other factors. Other scholars propose fuzzy (analytic hierarchy process) AHP-outranking frameworks for the prioritization of measures focused on the protection of water resources including water supply sources [19].

The advantages and disadvantages of different water resource allocation schemes were compared using a comprehensive evaluation and analysis method. Typical water resource evaluation methods include principal component analysis [20], projection pursuit [21], fuzzy comprehensive evaluation, and set pair analysis methods [22,23]. Compared with previous evaluation methods, set pair analysis can reflect the identity, difference, and opposition between evaluation indexes and standards. A pair formed by two sets that are connected in a certain way is called a set pair. If the two sets have some of the same characteristics, this is called an identity connection. If two sets have some opposite characteristics, this is called an opposition connection. If the connection between the two sets is neither an identity connection nor an opposite connection, then this is called a difference connection. Of these, identity connection and opposition connection are called deterministic connections of two sets, and the difference connection is called an uncertainty connection. The connection degree of the identity connection, difference connection, and opposition connection between the two sets can be expressed by the connection degree. Therefore, the evaluation dimension is more abundant and can display the influence of evaluation indexes on samples more comprehensively. Rich adjoint functions have been developed for the connection number of set pair analysis, including the subtractive set pair potential [24], partial connection number [25], and connection entropy [26,27,28].

Connection entropy was first proposed by Zhao on the basis of set pair analysis in 1992 [29]. It is further derived by calculating the connection component of the connection number and is an important adjoint function of the connection number. Connection entropy is applicable to measure the order and disorder of events, reflecting the internal disorder of set pair events [26,27]. Connection entropy can reflect the connection degree of set-pair events through identity entropy, opposition entropy, and difference entropy, which represent the entropy of the identity degree, opposition degree, and the difference degree between the evaluation samples and the evaluation standards. As a method combining set pair analysis and entropy, connection entropy can deal with system uncertainty as well as reflect the ordered development state in the system [29]. The identity entropy is equal to the information entropy, which can be used to measure the uncertainty degree of the system; the opposition can be used to measure the certainty degree in the system; and the difference entropy can be used to measure the part of the system where certainty and uncertainty coexist. It can also be said that the identity entropy is equal to the measurement of the disordered part, the opposition is equal to the measurement of the ordered part, and the different entropy is equal to the measurement of the intermediate state between ordered and disordered. Therefore, connection entropy is a kind of complete entropy, which can reflect the difference of the system in different states. In summary, connection entropy has unique advantages for the treatment and evaluation of water resource systems [26,27,28,29]. Therefore, the connection entropy method is selected in this paper. Because connection entropy cannot clearly reflect the physical connotation of identity, difference, and opposition entropy, this paper adopts a method for improving the connection entropy. We first calculate the samples’ connection numbers, then determine the difference coefficient, calculate the connection degree value of each indicator, and finally obtain the total entropy value of the connection number of each water resource allocation scheme. The optimal water resource allocation scheme can be obtained by comparing the total entropy of the connection number.

This study explores sustainable groundwater use by taking Beijing’s Daxing District as an example, collecting surface water and groundwater data in the study area from 2012 to 2016, and analyzing the applicability of water quality. According to the analysis results of water quality applicability and water supplies with different qualities, three groundwater exploitation modes were set up, and the water resources were optimally allocated under the three groundwater exploitation modes. Finally, the water resource allocation schemes were optimized based on the improved connection entropy. This study’s research methods and outcomes can provide methodological and theoretical lessons for water management in freshwater-deficient areas.

## 2. Materials and Methods

### 2.1. Study Area

The Daxing District is in Beijing’s southern plain (116°13′–116°43′ E and 39°26′–39°51′ N), and its administrative area comprises 14 township areas (Figure 1) of approximately 1036 km^2^. The study area has a warm, temperate, semi-humid continental monsoon climate, with an average annual rainfall of 510.1 mm. The annual rainfall distribution is uneven, and the interannual variability is large. The area belongs to the Yongding River alluvial plain with a flat terrain and a slope of 0.5–2.0‰. The regional surface water is derived from the region, and there are five rivers: the Xinfeng, Feng, Xiaolong, Tiantang, and Dalong Rivers. Figure 1 shows the spatial distribution of the rivers.

### 2.2. Data Sources

#### 2.2.1. Water Quality Data

This study uses surface water quality data of the five primary rivers and nine river sections in the Daxing District from 2012 to 2016, as shown in Figure 1. The Xinfeng, Feng, Dalong, and Tiantang Rivers involve two sections, and the Xiaolong River involves one section. The evaluation indexes of surface water quality primarily include NH_3_–N and COD. Twenty-five shallow groundwater monitoring wells’ data in the Daxing District from 2012 to 2016 were used (Figure 1). The groundwater quality evaluation index is NO_3_–N.

#### 2.2.2. Statistical Annual Data

This study’s statistical annual data include the Beijing Water Statistics Yearbook from 2012 to 2017, the Daxing Statistical Yearbook from 2005 to 2017, data compilation of the third agricultural census in the Daxing District, Beijing in 2016, the Bulletin of the Second National Pollution Source Census of Beijing in 2017, the China Environmental Statistical Yearbook in 2020, comprehensive planning of water resources in Daxing District of Beijing in 2018, and water supply planning for the river landscape in the Xinfeng River Basin of the Daxing District in 2017.

### 2.3. Research Method

This study compared surface water and groundwater quality data from 2012 to 2016 with the existing water standards to determine their applicability. According to water supplies with different qualities, the multiobjective optimization method was used to optimize the allocation of existing water resources, taking 2016 as an example. By constructing the evaluation system of water resource allocation schemes, the improved connection entropy was used to evaluate and analyze each water resource allocation scheme to select the best one.

#### 2.3.1. Multiobjective Optimal Water Resource Allocation Model

Objective Function

Sustainable water resource use is the premise of water resource allocation [30]; therefore, this study takes the minimum total water shortage and the minimum water cost as the objective functions. Water quality as a constraint condition is taken into account in the optimization model.

(1) Minimum total water shortage

The size or degree of water shortage has different degrees of influence on the development of all aspects of society; therefore, the minimum total water shortage is used as an objective function.
(1)$$f_{1}\left(q\right)=\min\sum_{j=1}^{4}\left[R_{j}-\sum_{i=1}^{4}q_{ij}\right]$$

In the formula, *f*_1_(*q*) is the total amount of water shortage (10^4^ m^3^), *q_ij_* is the amount of water supplied by the *i*th water source to the *j*th type water users (10^4^ m^3^), and *R^j^* is the water demand of the *j*th type users (10^4^ m^3^). *i* represents the *i*th water source, and *i* = 1, 2, 3, and 4 represent surface water, groundwater, reclaimed water, and transferred water, respectively. *j* represents the *j*th water user, and *j* = 1, 2, 3, and 4 represent domestic water, industrial water, agricultural water, and ecological water, respectively.

(2) Minimum water cost

The cost of using water determines the benefits of using water [31]. Depending on the total income, the lower the cost of using water, the greater the corresponding benefit. Therefore, the minimum using water cost is used as an objective function.
(2)$$f_{2}\left(q\right)=\min\sum_{i=1}^{4}\left(\sum_{j=1}^{4}q_{ij}\cdot p_{i}\right)$$

In the formula, *f*_2_(*q*) is the cost of using water (10^4^ yuan), and *p_i_* is the unit price of the water supply of the *i*th water source (yuan/m^3^). The unit price of water supply for groundwater and reclaimed water is determined according to the current supply unit price in Beijing, which is 5.00 yuan/m^3^ for groundwater and 1.00 yuan/m^3^ for reclaimed water. The unit price of transferred water is determined according to the water supply unit price in “the overall planning of the supporting project of the South-to-North Water Diversion Project in Beijing” (7.44 yuan/m^3^). Due to the poor water quality and limited application scope of surface water in southern Beijing, the surface water is calculated according to the existing agricultural water unit price (0.48 yuan/m^3^).

Constraint conditions

(1) Water supply capacity constraints

(3)$$\sum_{i=1}^{4}\sum_{j=1}^{4}q_{ij}\leq Q_{z}$$(4)$$\sum_{j=1}^{4}q_{ij}\leq Q_{i}$$
where *Q_z_* is the total amount of water available for each water source (10^4^ m^3^), and *Q_i_* is the available water amount of different water sources (10^4^ m^3^). According to the “comprehensive planning of water resources in Daxing District of Beijing”, the maximum available surface water (*Q*_1_) volume in the study area is 15.72 million m^3^, the maximum available groundwater (*Q*_2_) volume is 185.24 million m^3^, and the maximum available transferred water (*Q*_3_) volume is 26 million m^3^. According to the “water supply planning for river landscape in Xinfeng River Basin of Daxing District”, the maximum available reclaimed water (*Q*_4_) volume in the study area is 159.87 million m^3^.

(2) Water consumption constraint

Different water departments must meet maximum and minimum water requirements. The water consumption constraints are as follows:(5)$$Q_{\min,j}\leq Q_{j}\leq Q_{\max,j}$$
where *Q*_min,*j*_ and *Q*_max_,*j* represent the minimum and maximum water demand of the *j*th water user *Q_j_*, respectively. The minimum water demand, *Q*_min,1_ and *Q*_min,2_, for domestic and industrial water is 95%, and the maximum value is 110%. The minimum water demand for agricultural and ecological water is 90% for *Q*_min,3_ and *Q*_min,4_, and the maximum water demand is 110%. Furthermore, the water supply and demand in the above formula should meet the nonnegative constraints.

#### 2.3.2. Model Solution Based on Algorithm Preferences

The constructed water resource optimization model belongs to the multiobjective optimization problem with constraints. The traditional particle swarm and ant colony algorithms easily become local when dealing with multiobjective problems, and the Pareto solution’s convergence effect is poor [32,33]. Therefore, this study selects three algorithms with good performance when dealing with constrained multiobjective optimization problems to solve the model [34,35,36]: the nondominated sorting genetic algorithm-II (NSGA-II), nondominated sorting genetic algorithm-III (NSGA-III), and a multiobjective evolutionary algorithm based on decomposition (MOEA/D). 

Deb proposed the NSGA-II algorithm in 2002, mainly to improve the shortcomings of the NSGA algorithm [34]. Compared with NSGA, NSGA-II adopts a fast non-dominated sorting method, which greatly reduces the computation time. The elite strategy ensures that excellent individuals can be retained with greater probability. The crowding method was used instead of the fitness sharing strategy, which was necessary to ensure the diversity of individuals. The algorithm frameworks of NSGA-III and NSGA-II are roughly the same, but the selection mechanism is different [35]. NSGA-II uses crowding to select individuals with the same non-dominated level, while NSGA-III selects individuals based on reference points. NSGA-III adopts the method based on reference points to solve the problem of poor convergence and the diversity of the algorithm if the crowding distance is continued in the face of multi-objective optimization problems with three or more objectives.

In order to find the Pareto optimal solution of the multiobjective optimization problem, Zhang proposed MOEA/D [36]. The core idea of this method is to transform the multiobjective optimization problem into a series of single-objective optimization sub-problems or multiple multiobjective sub-problems and then use the neighborhood relationship between sub-problems to optimize these sub-problems in a collaborative manner to approach the entire Pareto front. Usually, the definition of sub-problems is determined by the weight vector, and the neighborhood relationship between subproblems is determined by calculating the Euclidean distance between the weight vectors.

The evaluation using the hypervolume (*HV*) metric is preferred for the solved results of the algorithms [37,38]. The larger the *HV* value, the better the algorithm’s overall performance. The formula for calculating the *HV* index is as follows:(6)$$HV=\sigma \cdot U_{c=1}^{\left|s\right|}\left(v_{c}\right)$$
where *HV* represents the calculation result of the *HV* index. *σ* represents the Lebesgue measure for measuring volume. $\left|s\right|$ is the number of nondominated solution sets, and *v_c_* is the *HV* formed by the reference point and the *c*th solution in the Pareto solution set.

#### 2.3.3. Optimized Selection of Water Resource Allocation Schemes Based on the Improved Connection Entropy Method

To compare the advantages and disadvantages of different water resource allocation schemes, the total entropy of the connection number of each water resource allocation scheme is calculated based on the improved connection entropy method. The optimal water resource allocation scheme can be obtained by comparing the total entropy of the connection number.

The connection entropy is a method that combines set pair analysis and entropy. Therefore, connection entropy has the characteristics of set pair analysis to deal with uncertain problems in the system, and it can also reflect the orderly development of the system [29].

Regarding the descriptions of uncertain systems, one is the probability and statistics theory to describe random uncertainty, and the other is the fuzzy mathematical theory to describe fuzzy uncertainty. The theory of probability and statistics overemphasizes the independence of the system, while the theory of fuzzy mathematics relies too much on subjective experience, so both theories have shortcomings. Set pair analysis, first proposed by Zhao in 1989, is commonly used to deal with random uncertainty and fuzzy uncertainty problems [29,39,40]. When dealing with uncertainty, it is more objective, and the operation is simpler. The approach works by establishing the connection number between samples and rules or levels to evaluate the system, which can be evaluated qualitatively or quantitatively, meaning that the evaluation results are more objective.

The general expression of the connection number is as follows (7):(7)$$u=a+bI+cJ=u_{1}+u_{2}+u_{3}$$
where *u* denotes the connection degree between two set pair events, also known as the ternary connection number. *a*, *b* and *c* are the degree of identity, difference, and opposition of two set pair events, respectively, and the ranges of *a*, *b,* and *c* are [0, 1], and *a* + *b* + *c* = 1; *I* represents the difference coefficient, and the value interval is [−1, 1]; *J* denotes the opposition coefficient, *J* = −1. *u*_1_, *u*_2_ and *u*_3_ represent the identity degree component, difference degree component, and opposition degree component of the connection degree *u*, respectively. 

The connection degree of the identity connection, difference connection, and opposition connection between the two sets can be expressed by connection degree *u*. The expression *a* + *bI* + *cJ* of connection degree is the connection number of the sample.

Establishing a comprehensive evaluation model based on the improved contact entropy includes the following steps.

Step 1: Establish an evaluation index system and calculate the index weight

Because the water resource allocation scheme will affect all aspects of social development, this study constructs an evaluation index system according to the principles of applicability, operability, and comprehensiveness, considering various factors such as regional natural environment conditions and economic development [41]. The evaluation index system includes three subsystems and nine indicators. The three subsystems involve social, economic, and ecological benefits. Social benefits include per capita water quota, agricultural water-use ratio, and water consumption of 10,000 yuan industrial output value. Economic benefits include the total cost of water supply, the cost of groundwater supply, the cost of reclaimed water, and the proportion of reclaimed water. The ecological benefits include the rate of groundwater development and the maximum concentration of NO_3_–N in groundwater. 

The analytic hierarchy process (AHP) is a subjective weighting method [42], and its advantage lies in the fact that the weight can be assigned by subjective judgment under the condition of insufficient sample data, so this method is not very ideal in terms of credibility. The entropy weight method is an objective weighting method [28], which makes full use of the original data information entropy. The reliability is ideal under the condition of high completeness of the sample data, but it has a slight disadvantage in reflecting the knowledge and experience of experts and the opinions of decision makers. Therefore, by combining the advantages and disadvantages of these two methods, this paper uses the combination weighting method of the entropy weight method and analytic hierarchy process to determine the index weight. 

The entropy weight method and AHP determine each indicator’s weight [43] and can be expressed as
(8)$$w_{j}=\beta w_{j}^{\prime }+(1-\beta )w_{j}^{\prime \prime }$$
where $w_{j}$ is the combined weight value of the *j*th index, and $w_{j}^{\prime }$ and $w_{j}^{\prime \prime }$ are the weight values of entropy weight method and analytic hierarchy process, respectively. *β* is the preference coefficient, taking 0.5.

The evaluation indicators and indicator weights of the optimal water resource allocation scheme are shown in Table 1.

Step 2: Determine the evaluation index level

Based on the comprehensive consideration of the social, economic, and ecological environments and other factors in the study area [44] and referring to the existing data [39,42,45], the evaluation grade standard of optimal water resource allocation is established as follows:(9)$$\left\{\left.s_{gj}\right|\right.g=1,2,3;j=1,2,3,\cdots ,\left.n\right\}$$
where *S_gi_* is the level limit for each evaluation indicator. *g* is the evaluation level—levels I, II, and III. *j* represents the *j*th indicator in sample *i*, and *n* is the number of indicators. The level limit for each evaluation indicator is shown in Table 1.

The sample data set is expressed as
(10)$$\left\{\left.x_{ij}\right|\right.i=1,2,3,\cdots ,m;j=1,2,3,\cdots ,\left.n\right\}$$
where *x_ij_* represents each sample’s index value, *i* is the evaluation samples, and *m* is the number of samples.

Step 3: Calculate the connection number of each evaluation index

Calculate the index connection number between the *j*th index value of the evaluation sample *i* and the evaluation standard of optimal water resource allocation. If the indicator is positive with *x_ij_* ≤ *s*_1*j*_, or negative with *x_ij_* ≥ *s*_1*j*_,
(11)$$u_{ij1}=\left\{\begin{array}{l} 1\\ 1-2\frac{x_{ij}-s_{1j}}{s_{2j}-s_{1j}}\\ -1 \end{array}\right.$$

If the indicator is positive with *s*_1*j*_ < *x_ij_* ≤ *s*_2*j*_, or negative with *s*_1*j*_ > *x_ij_* ≥ *s*_2*j*_,
(12)$$u_{ij2}=\left\{\begin{array}{l} 1-2\frac{s_{1j}-x_{ij}}{s_{1j}-s_{0j}}\\ 1\\ 1-2\frac{x_{ij}-s_{2j}}{s_{3j}-s_{2j}} \end{array}\right..$$

If the indicator is positive with *s*_2*j*_ < *x_ij_* ≤ *s*_3*j*_, or negative with *s*_2*j*_ > *x_ij_* ≥ *s*_3*j*_*,*
(13)$$u_{ij3}=\left\{\begin{array}{l} -1\\ 1-2\frac{s_{2j}-x_{ij}}{s_{2j}-s_{1j}}\\ 1 \end{array}\right..$$
where *u_ij_*_1_, *u_ij_*_2_, and *u_ij_*_3_, respectively, represent the index connection number of the *j*th index of sample *i* in various levels. *s*_1*j*_ and *s*_2*j*_ are the limit values of level 1 and level 2, level 2, and level 3, respectively. *s*_0*j*_ and *s*_3*j*_ are the minimum values of level 1 and the maximum value of level 3, respectively.

The identity, difference, and oppositeness degrees of the connection number of the *j*th index of sample *i* are converted into relative membership degrees. The calculation process is as follows:(14)$$v_{ijg}^{\prime }=0.5+0.5u_{ijg}$$
(15)$$v_{ijg}={v^{\prime }}_{ijg}/\sum_{g=1}^{3}{v^{\prime }}_{ijg},$$
(16)$$u_{ij}=v_{ij1}+v_{ij2}I+v_{ij3}J.$$
where *u_ij_* represents each indicator’s connection degree. *v_ij_*_1_$v_{ij1}v_{ij2}v_{ij3}$, *v_ij_*_2_, and *v_ij_*_3_, respectively, represent the identity, difference, and oppositeness degrees of the *j*th index of sample *i*, and *v_ijg_*$v_{ijg}^{\prime }$ is the calculation process quantity.

Step 4: Calculate each indicator’s connection value

The difference degree coefficient is determined using the proportional value method. The difference degree *v_ij_*_2_*I* in Formula (16) is taken as $v_{ij1}\cdot v_{ij2}-v_{ij3}\cdot v_{ij2}$. The opposite degree coefficient *J* = −1, and the connection value *u_ij_* of each evaluation index value is calculated.

Step 5: Calculate each indicator’s connection entropy

The improved connection entropy comprises identity, difference, and opposition entropy [26]. The identity entropy, difference entropy, and opposition entropy represent the entropy of the identity degree, opposition degree, and the difference degree between the evaluation samples and the evaluation standards of water resource allocation schemes, respectively. The formulas for calculating each entropy and the total entropy are as follows:(17)$$S_{a}=-\sum w_{j}In(v_{ij1})$$
(18)$$S_{b}=-\sum w_{j}In(v_{ij2})$$
(19)$$S_{c}=-\sum w_{j}In(v_{ij3})$$
(20)$$S=-\sum w_{j}In(0.5u_{ij}+0.5)$$
where *S*, *S_a_*, *S_b_*, and *S_c_* represent the total, identity, difference, and opposition entropy of the *j*th index of sample *i*, respectively. *W_j_* is the weight of the *j*th indicator.

## 3. Results and Discussion

### 3.1. Applicability Evaluation of Water Quality

By collecting the water quality data of surface water and groundwater in the study area from 2012 to 2016, their applicability was evaluated and analyzed. Surface water quality evaluation involved NH_3_–N and COD, and groundwater quality involved the NO_3_–N index.

#### 3.1.1. Applicability Evaluation of Surface Water Quality

According to environmental quality standards (GB 3838-2002), surface water is divided into classes I–V. Water inferior to category V is further classified as category V_1_, V_2_, V_3_, and V_4_ according to the discharge standard of pollutants for municipal wastewater treatment plants (GB 18918-2002). Table 2 shows the division criteria of index limits corresponding to NH_3_–N and COD.

Figure 2a shows that the NH_3_–N index in surface water was in the range of V_1_–V_4_ from 2012 to 2016, and the water quality was poor. The proportions of category V_1_, V_2_, V_3_, and V_4_ water samples were 6.67%, 28.88%, 26.67%, and 37.78%, respectively, showing that V_4_ water accounts for the largest proportion. From 2012 to 2016, the NH_3_–N pollution shows a weakening trend, from 41.80 mg/L in 2012 to 14.46 mg/L in 2016. However, the NH_3_–N index was still in the inferior class V, exceeding the class V standard limit of 2.00 mg/L.

According to the NH_3_–N pollution spatial distribution in surface water, the Xiaolong River had the highest pollution degree. The average NH_3_–N index from 2012 to 2016 reached 33.70 mg/L, followed by the Xinfeng River (31.72 mg/L), Tiantang River (27.57 mg/L), Dalong River (20.57 mg/L), and the Feng River (16.76 mg/L) with the lowest pollution degree.

Figure 2b shows that the COD index in surface water was in the range of category III–V_4_, in which the proportion of III–V was 15.56%, and the proportions of categories V_1_, V_2_, V_3_, and V_4_ of water samples were 8.89%, 24.44%, 24.44%, and 26.67%, respectively. The proportion of V_4_ water was the largest.

The COD index change from 2012 to 2016 showed a weakening trend. From 2012 to 2013, the COD value showed an upward trend, from 65.04 mg/L to 153.39 mg/L, indicating that COD pollution emissions in the study area increased. From 2013 to 2016, the COD value decreased from 153.39 mg/L to 58.00 mg/L.

The water quality of surface water was compared with the water quality standards of various water uses to evaluate the applicable scope of surface water. Table 3 shows the evaluation results of water quality applicability. From Table 3, surface water could not be used for domestic, industrial, and landscape environment water but could be used for agricultural irrigation water. According to the comprehensive water resource planning in Beijing’s Daxing District (2018), the surface water resources in the study area were 29.64 million m^3^. Because the rainfall from June to September accounts for ~82% of the annual rainfall, the surface water was used as the surface water source. Due to the lack of extensive rainfall collection facilities in the study area, the actual available water was 15.72 million m^3^, according to the existing sluice dam and pond water storage capacity.

The NH_3_–N and COD surface water pollution was severe. To further analyze the surface water pollution sources, the pollution sources of NH_3_–N and COD in surface water from 2012 to 2016 were analyzed according to the second national pollution source census bulletin in Beijing.

According to the second national pollution source census bulletin in Beijing, the COD emission in Beijing was 92.10 million kg, among which the proportions of industrial, agricultural, domestic, and centralized pollution facilities were 1.61%, 28.87%, 69.35%, and 0.17%, respectively. The NH_3_–N emission was 3.44 million kg, and the proportions of industrial, agricultural, domestic, and centralized treatment facilities were 1.28%, 12.17%, 85.90%, and 0.65%, respectively. According to the actual water consumption in Beijing in 2017, NH_3_–N and COD emissions per unit of water consumption of industrial, agricultural, and domestic water could be calculated. According to the water consumption from 2012 to 2016, the pollutant emission ratios of various water consumptions in the study area could be calculated. Figure 2 shows the river’s NH_3_–N and COD concentrations from 2012 to 2016. In the case of excluding the background values of NH_3_–N and COD, the total pollutant emissions in the river could be obtained according to the annual water supplement. Finally, the contribution rates of industrial, agricultural, and domestic sources to river pollutant emissions were obtained. Figure 3 shows the results.

Figure 3a shows that the contribution of various sources to NH_3_–N pollution had a downward trend, and the largest decrease was agricultural and domestic sources, which decreased by 4.75 × 10^5^ kg and 3.98 × 10^5^ kg, respectively. The average contribution rate of each source to NH_3_–N pollution was domestic (55.28%) > agricultural (43.29%) > industrial (0.78%) > other anthropogenic (0.65%) pollution sources.

Figure 3b shows that the contribution of various sources to COD pollution also showed a downward trend overall. From 2012 to 2013, the total contribution of pollutants showed an upward trend, and from 2013 to 2016, this decreased significantly. The largest fluctuation of COD contribution was the agricultural pollution source, probably related to agricultural breeding. According to the Daxing District statistical yearbook, from 2012 to 2013, the number of live pigs increased from 2.55 × 10^5^ to 2.64 × 10^5^, and from 2013 to 2016, it decreased from 2.64 × 10^5^ to 1.81 × 10^5^, verifying that the pollution sources of COD were from agricultural breeding. The average contribution rate of each source to COD pollution was agricultural (68.77%) > domestic (30.39%) > industrial (0.67%) > other anthropogenic (0.17%) pollution sources.

#### 3.1.2. Applicability Evaluation of Groundwater Quality

Groundwater is the primary water supply source in the study area, and its water quality is critical to the region’s stability and development [46,47]. Because NO_3_–N pollution was the primary groundwater pollutant in Beijing [45,47], the pollution range accounts for 5% of the total area. Simultaneously, NO_3_–N, as a toxicological index, was affected by human activities considerably and had the greatest impact on human health and livestock hazards [48,49]. Therefore, NO_3_–N was selected as an evaluation index in this paper.

By collecting groundwater’s NO_3_–N water quality data in the study area from 2012 to 2016, the water quality samples were classified according to the groundwater quality standard (GB/T 14848-2017). Figure 4a shows the classification results. From 2012 to 2016, there were 69 samples of class I water (55.20%), 17 samples of class II water (13.60%), 33 samples of class III water (26.40%), 5 samples of class IV water (4.00%), and 1 sample of class V water (0.80%). Thus, the groundwater quality was good, primarily class I–III water, accounting for 95.20% of the total samples.

The NO_3_–N concentrations continuously increased from 2012 to 2016, increasing slowly from 4.05 mg/L to 4.99 mg/L. From a spatial viewpoint (Figure 4b), the NO_3_–N concentrations in the northern region were the largest, with an average concentration of 13.92 mg/L, followed by the central region and the smallest in the southern region. The most polluted area in the northern region was Jiugong–Yinghai–Yizhuang. From 2012 to 2016, the average NO_3_–N concentration reached 24.56 mg/L.

Because the water quality standards of industrial, agricultural, and ecological water referred to in this paper (Table 3) did not involve the limit value of NO_3_–N but only domestic water, it could be considered that groundwater met the requirements of industrial, agricultural, and ecological water. Simultaneously, it was necessary to evaluate further the applicability of the NO_3_–N index to domestic water.

To further evaluate the applicability of groundwater to domestic water, it was necessary to combine the urban water supply quality standard (CJ/T 206-2005) and the drinking water hygiene standard (GB 5749-2006). According to the drinking water hygiene standard (GB 5749-2006), the limit value of NO_3_–N in groundwater was 20 mg/L, and according to the urban water supply quality standard (CJ/T 206-2005), it was 10 mg/L (20 mg/L in exceptional cases). Therefore, in the case of sufficient water, NO_3_–N concentrations of 10–20 mg/L should not be used as domestic water. During severe water shortage, the areas with NO_3_–N concentrations of 10–20 mg/L could be used as domestic water after treatment, and in areas where the concentration was 20 mg/L, the water should not be used as domestic water.

Because groundwater was buried deep underground, the groundwater pollution treatment was more challenging than surface water. Therefore, to ensure social water use, areas with severe groundwater quality pollution were gradually reduced after self-purification and fresh rainwater supplement by reducing or prohibiting the exploitation amount to achieve the minimum cost of treatment and the best effect of groundwater pollution control.

### 3.2. Optimized Selection of Algorithms of Water Resource Allocation Schemes

Since the geological conditions of the study area are relatively uniform, the mining wells mainly exploit and utilize quaternary pore water, and the water consumption structure and water supply sources of the study area are relatively stable for many years without significant change. 2016 was selected as a typical year for optimization, which can reflect the basic situation of the study area.

Taking 2016 as an example, three groundwater exploitation modes were set up to optimize water resource allocation in the study area.

Groundwater exploitation mode 1: The areas where groundwater NO_3_–N concentrations were above 20 mg/L were not regarded as domestic water. 

As can be seen from Figure 4b, the areas with NO_3_-N concentrations above 20 mg/L were mainly located in Jiugong, Yinghai, and Yizhuang, covering an area of 35.67 km^2^. The area with NO_3_–N above 20 mg/L account for 59.92%, 30.35%, and 19.34% of the total area of Jiugong, Yinghai, and Yizhuang, respectively. According to the actual domestic water consumption in 2016 and the above area ratio, it can be calculated that the groundwater exploitation volumes of the area with NO_3_–N concentrations above 20 mg/L in Jiugong, Yinghai, and Yizhuang were 5.00 million m^3^, 1.10 million m^3^, and 0.53 million m^3^, respectively, totaling 6.63 million m^3^. Therefore, according to groundwater exploitation mode 1, the available groundwater supply decreased by 6.63 million m^3^, and the available groundwater supply was 178.61 million m^3^.

Groundwater exploitation mode 2: Groundwater in areas with NO_3_–N concentrations of 10–20 mg/L was not used as domestic water, and groundwater exploitation in areas with NO_3_–N concentrations above 20 mg/L was prohibited. 

Figure 4b shows that the areas with NO_3_–N concentrations between 10 mg/L and 20 mg/L were mainly located in Huangcun, Yinghai, Xihongmen, Jiugong, and Yizhuang, covering an area of 166.77 km^2^. In addition, the area of NO_3_–N concentrations of 10~20 mg/L accounted for 59.81%, 30.69%, 100%, 40.08%, and 80.66% of each of the above townships’ total area, respectively. According to the actual domestic water consumption in Huangcun, Yinghai, Xihongmen, Jiugong, and Yizhuang in 2016, those areas’ groundwater exploitation of NO_3_–N concentrations of 10~20 mg/L were 9.09 million m^3^, 1.11 million m^3^, 5.52 million m^3^, 3.34 millon m^3^, and 2.23 million m^3^—a total of 21.29 million m^3^.

From the above, the area of NO_3_–N concentrations above 20 mg/L accounted for 59.92%, 30.35%, and 19.34% of the total area of Jiugong, Yinghai, and Yizhuang. According to the actual groundwater exploitation of Jiugong, Yinghai, and Yizhuang in 2016, the groundwater exploitation of those areas with NO_3_–N above 20 mg/L were 5.78 million m^3^, 1.75 million m^3^, and 0.71 million m^3^—a total of 8.24 million m^3^ by the above area ratio.

Therefore, according to groundwater exploitation mode 2, the available groundwater water supply decreased by 29.53 million m^3^, and the available groundwater water supply is 155.71 million m^3^.

Groundwater exploitation mode 3: Groundwater exploitation was prohibited in areas where NO_3_–N concentrations were above 10 mg/L. 

Figure 4b shows that the areas with NO_3_–N above 10 mg/L were mainly located in Huangcun, Yinghai, Xihongmen, Jiugong, and Yizhuang, and the total area was 202.44 km^2^. The area with NO_3_–N above 10 mg/L accounted for 59.81%, 61.04%, 100%, 100%, and 100% of Huangcun, Yinghai, Xihongmen, Jiugong, and Yizhuang, respectively. According to the actual groundwater exploitation of Huangcun, Yinghai, Xihongmen, Jiugong, and Yizhuang in 2016 and the above area ratio, the groundwater exploitation rates of Huangcun, Yinghai, Xihongmen, Jiugong, and Yizhuang with NO_3_–N concentrations above 10 mg/L were 17.98 million m^3^, 3.52 million m^3^, 7.95 million m^3^, 9.64 million m^3^, and 3.68 million m^3^—a total of 42.77 million m^3^. Therefore, according to groundwater exploitation mode 3, the available groundwater water supply decreased by 42.77 million m^3^, and the available groundwater water supply was 142.47 million m^3^.

In this paper, two objective functions were set up: the minimum total water shortage and the minimum total water cost. The 10 decision variables included groundwater, reclaimed and transferred water consumed by domestic water, groundwater and reclaimed water consumed by industrial water, surface water, groundwater and reclaimed water used in agriculture, and groundwater and reclaimed water consumed by ecological water. NSGA-II, NSGA-III, and MOEA/D algorithms were used to solve the water resource allocation model under groundwater exploitation modes 1–3, and each algorithm’s mean HV and standard deviation of HV were obtained (Table 4).

Table 4 shows that NSGA-III showed the largest mean value and the smallest standard deviation of HV, indicating that NSGA-III had better performance in solving the established multiobjective optimization model compared with NSGA-II and MOEA/D. Compared with MOEA/D, the HV mean of NSGA-II was larger than that of MOEA/D, but the HV standard deviation of NSGA-II was larger than that of MOEA/D, indicating that the performance difference between NSGA-II and MOEA/D was small, but the performances of NSGA-II and MOEA/D were inferior to that of NSGA-III. Therefore, we chose NSGA-III as the solution algorithm of the optimization model in this paper.

Table 5 shows the actual water consumption in 2016 and the water resource allocation results under groundwater exploitation modes 1–3.

Table 5 shows that the actual water consumption structure in 2016 was unreasonable. (1) The use of surface water was 0. According to the above analysis, surface water in the study area could be used as agricultural water; therefore, surface water was wasted. (2) The available groundwater in the study area was 185.24 million m^3^, whereas the actual groundwater exploitation amount was 213.91 million m^3^, and the groundwater exploitation rate was 115.48%, which might cause the continuous deterioration of the groundwater level and environment. (3) Groundwater was used for livelihood, industry, and agriculture, reclaimed water was rarely used, and most reclaimed water was used only for ecology. According to the standard for water-saving design in civil building (GB 50555-2010), the toilet flushing ratio in civil residential buildings was 21%, and reclaimed water could be used for flushing toilets. However, the actual amount of reclaimed water for domestic use was less than 0.07%. (4) Compared with the local groundwater, surface, and reclaimed water, transferred water was more expensive. As a source of good water quality, transferred water should be preferentially used as domestic water. Some of the transferred water was used for industrial water, which was unreasonable.

Table 5 shows that for groundwater exploitation modes 1–3, the optimal water resource allocation schemes improved compared with the actual water consumption in 2016. (1) According to the applicability evaluation results of surface water quality in Section 3.1.1, 15.72 million m^3^ of surface water was used for agricultural irrigation, reducing surface water waste. (2) The groundwater extraction volumes of groundwater exploitation modes 1–3 were 148.97 million m^3^, 149.01 million m^3^, and 142.47 million m^3^, respectively, and the groundwater exploitation rates were 80.42%, 80.44%, and 76.91%, respectively, effectively alleviating the continuous deterioration of the groundwater environment (in order to compare the groundwater development rate of water resource allocation schemes easily, the rate of groundwater development refers to the ratio of the groundwater exploitation amount to the maximum available water volume, which is 185.24 million m^3^). (3) The water structure was further optimized. For groundwater exploitation modes 1–3, the guaranteed rate of domestic and industrial water reached 100%, and that of agricultural and ecological water reached more than 90%. The proportion of reclaimed water in domestic water was ~20%, and the proportion of reclaimed water in industrial water was more than 80%. (4) As a high-quality water source, transferred water was only used as domestic water. (5) For groundwater exploitation modes 1–3, the groundwater pollution was gradually reduced by limiting or prohibiting groundwater exploitation in areas with severe NO_3_–N pollution. The groundwater exploitation for modes 1–3 decreased by 6.63 million m^3^, 29.53 million m^3^, and 155.71 million m^3^, respectively. The amount of groundwater extraction was reduced, equivalent to supplementing the dilution water with lower NO_3_–N concentrations. After calculation, the maximum NO_3_–N concentrations of groundwater exploitation modes 1–3 were 25.28 mg/L, 16.49 mg/L, and 13.73 mg/L, respectively, which were reduced by 15.45%, 44.85%, and 54.08%, compared with the actual water consumption of 29.90 mg/L in 2016.

### 3.3. Optimized Selection of Water Resource Allocation Schemes

The statistical and actual water consumption data from 2006 to 2016 were collected to select the water resource allocation schemes. The water resource allocation schemes for three groundwater exploitation modes were compared with the actual water consumption from 2006 to 2016, and the best scheme was selected. Table 1 shows the optimal water resource allocation scheme’s evaluation indicators, standard grades, and indicator weights.

According to Formulas (8)–(20), the total entropy of the connection number of the actual water-use schemes from 2006 to 2016 and the optimal water resource allocation schemes for the three groundwater exploitation modes were calculated (see Appendix A). Appendix A show the connection number of indicators of the actual water-use schemes from 2006 to 2016 and the optimal water resources allocation schemes for the three groundwater exploitation modes. Appendix A shows the connection entropy of the actual water-use schemes from 2006 to 2016 and the optimal water resource allocation schemes for the three groundwater exploitation modes.

Figure 5 shows the calculation results.

Figure 5a shows that the entropy value of the actual water-use schemes’ total benefit from 2006 to 2016 decreased, from 1.40 in 2006 to 0.63 in 2016, indicating that the actual water-use schemes’ total benefit was growing. From 2006 to 2010, the total entropy of the actual water-use schemes fluctuated around 1.46. However, the total entropy decreased considerably from 1.64 in 2010 to 0.45 in 2013. On the one hand, due to the decrease in agricultural water consumption from 264 million m^3^ in 2010 to 215 million m^3^ in 2013, the proportion of agricultural water consumption decreased; therefore, the social benefit entropy was in a downward trend in this period—the social benefit entropy decreased from 0.40 in 2010 to 0.05. On the other hand, reclaimed water-use increased from 115 million m^3^ in 2010 to 119 million m^3^. Due to the low cost of reclaimed water, the economic benefit entropy also decreased during this period, from 0.94 in 2010 to 0.36. From 2013 to 2016, the entropy value of the total benefit increased first and then decreased slowly, primarily due to the increase in the entropy value of the ecological benefit. From 2013 to 2016, the study area’s population increased from 1.51 million to 1.69 million, increasing the domestic water consumption. The reclaimed water was primarily used as ecological water; therefore, the increase in domestic water consumption caused by population growth could only increase the groundwater exploitation, which increased the use rate of groundwater development. Secondly, the maximum NO_3_–N concentrations in groundwater increased from 18.90 mg/L in 2013 to 34.00 mg/L in 2015 and decreased to 29.90 mg/L in 2016. The maximum NO3–N concentration change was consistent with the total benefit entropy curve.

Figure 5b shows that the total benefit entropy values of the water resource optimization schemes for groundwater exploitation modes 1–3 were 0.58, 0.39, and 0.38, respectively—all less than the total benefit entropy value of the actual water-use scheme in 2016 (0.63). The water resource optimization schemes for groundwater exploitation modes 1–3 were better than the actual water-use scheme in 2016. The total benefit entropy value of groundwater exploitation mode 3 was smaller than modes 1 and 2 and smaller than the total benefit entropy value of the actual water-use schemes from 2006 to 2016. Therefore, groundwater exploitation mode 3 was better than groundwater exploitation modes 1 and 2 and better than the actual water-use schemes from 2006 to 2016. Under the premise of meeting the guaranteed water-use rate, the water resource allocation schemes for groundwater exploitation modes 1–3 and the actual water-use schemes in 2016 were 80.41%, 80.43%, 76.91%, and 115.48%, respectively. The maximum groundwater NO_3_–N concentrations were 25.28 mg/L, 16.49 mg/L, 13.73 mg/L, and 29.90 mg/L, respectively. Groundwater exploitation mode 3 protected the groundwater to the greatest extent and realized sustainable use compared to groundwater exploitation modes 1 and 2 and the actual water-use scheme in 2016. Therefore, it was recommended to use groundwater exploitation mode 3 to optimize the water resource allocation in the study area.

## 4. Conclusions

This study evaluated the applicability of Beijing’s Daxing District’s surface water and groundwater quality. Taking 2016 as an example, according to water supplies with different qualities, the existing water-use scheme was optimized under three groundwater exploitation modes. After optimization, the surface and reclaimed water in the study area had been fully used, and the development and use rate of groundwater and NO_3_–N pollution had been effectively improved. Therefore, the water resource optimization scheme of groundwater exploitation mode 3 was recommended. With the continuous improvement of urbanization, groundwater overexploitation and pollution have become increasingly prominent. Water resource managers can take the following measures to manage or mitigate these problems. (1) It is necessary to fully develop and use unconventional water sources, such as reclaimed water and rainwater, according to local conditions. Furthermore, deep confined water exploitation should be prohibited and water safety ensured according to water supplies with different qualities. (2) Domestic water should be based on the full use of transferred water, and an appropriate amount of groundwater could be exploited. Industrial water should rely on reclaimed water, ecological water should use reclaimed water and rainwater, and groundwater should not be used. (3) Domestic sewage and industrial wastewater should be fully collected and treated in a centralized manner before being discharged into surface water after reaching the standard. The use of agricultural fertilizers should be restricted or reduced.

The hydrological cycle is an open system with great uncertainty; especially with the intensification of climate change and the impact of human activities, the uncertainty in the process of the hydrological cycle is greatly increased. For example, the so-called Hurst phenomenon has been shown to be one of the main factors causing severe droughts and water shortages, resulting in groundwater overexploitation. Entropy is regarded as a measure of system disorder and uncertainty. Although it has been widely used in hydrology, due to the limitation of the theory and method and the high complexity of hthe ydrological system, it is necessary to further study the entropy theory to solve hydrological problems. As a kind of complete entropy, connection entropy has great research value in dealing with complex hydrological problems.

## Figures and Tables

**Figure 1 entropy-24-00920-f001:**
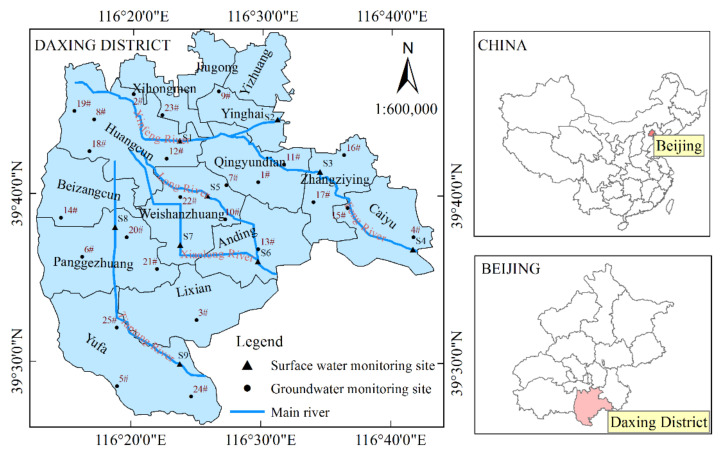
Geographic map of the study area.

**Figure 2 entropy-24-00920-f002:**
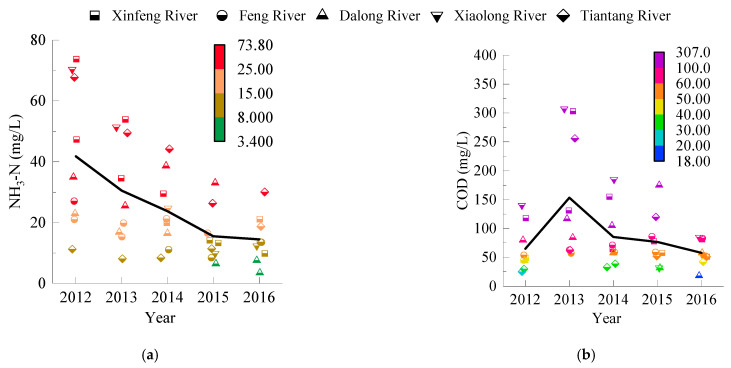
Water quality of NH_3_–N and COD in the study area’s surface water. (**a**) Water quality of NH_3_–N in the study area’s surface water. (**b**) Water quality of COD in the study area’s surface water.

**Figure 3 entropy-24-00920-f003:**
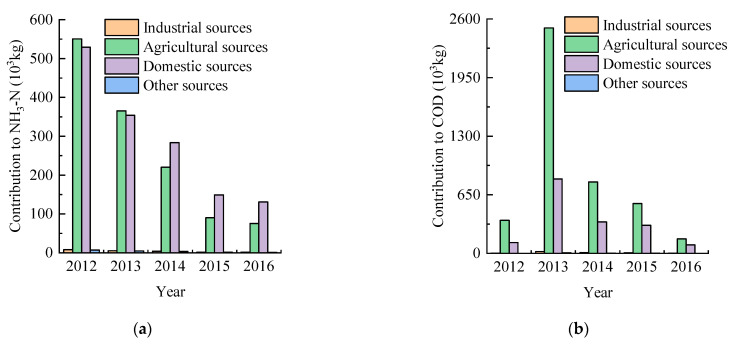
The contribution rate of various pollution sources to river pollution. (**a**) The contribution of various sources to NH_3_–N pollution. (**b**) The contribution of various sources to COD pollution.

**Figure 4 entropy-24-00920-f004:**
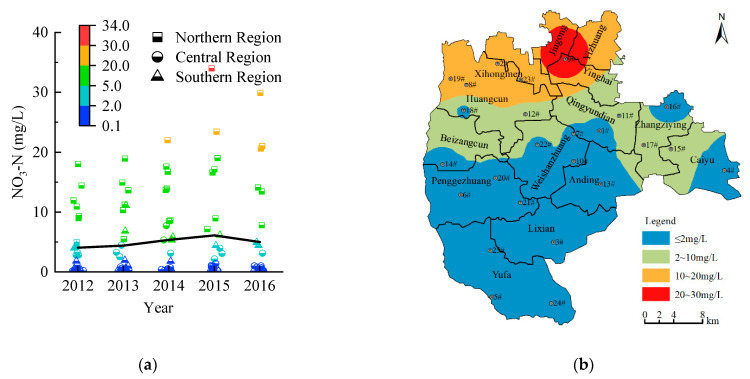
Interval distribution of NO_3_–N concentration in groundwater at each monitoring point and spatial distribution of average NO_3_–N concentration from 2012 to 2016. (**a**) Interval distribution of NO_3_–N concentration in groundwater at each monitoring point from 2012 to 2016. (**b**) Spatial distribution of average NO_3_–N concentration from 2012 to 2016.

**Figure 5 entropy-24-00920-f005:**
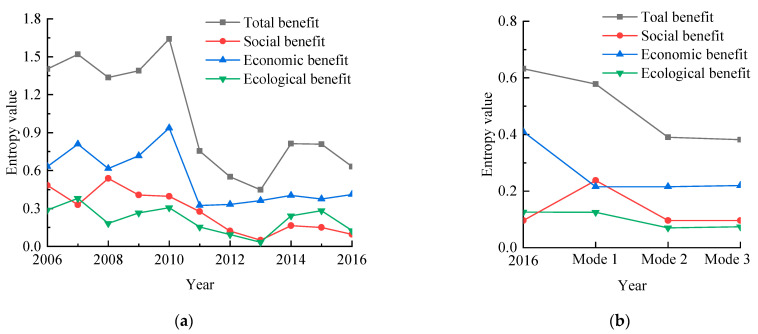
Comparison of entropy values between the actual water-use schemes from 2006 to 2016 and the optimal water resource allocation schemes for the three groundwater exploitation modes. (**a**) Entropy values of the actual water-use schemes from 2006 to 2016. (**b**) Entropy values of the optimal water resource allocation schemes for the three groundwater exploitation modes.

**Table 1 entropy-24-00920-t001:** Evaluation indicators, standard grades, and indicator weights of the optimal water resource allocation scheme.

Serial Number	Target	Evaluation Indicators	Symbol	Indicator Level			Weight
				Good (Level I)	Medium (Level II)	Poor (Level III)	
1	Social benefit	Water quota per capita (L/(person∙day))	X1	>154.83	[146.71, 154.83]	<146.71	0.1174
2		Agricultural water-use ratio	X2	<0.36	[0.36, 0.72]	>0.72	0.0981
3		Water consumption per 10,000 yuan of industrial output value (m^3^/10^4^ yuan)	X3	<2.78	[2.78, 4.15]	>4.15	0.1010
4	Economic benefit	Total cost of water supply (100 million yuan)	X4	<13.66	[13.66, 14.42]	>14.42	0.1036
5		Groundwater supply cost (100 million yuan)	X5	<10.69	[10.69, 11.05]	>11.05	0.0970
6		Reclaimed water supply cost (100 million yuan)	X6	>1.24	[1.14, 1.24]	<1.14	0.1398
7		Reclaimed water supply ratio	X7	>0.34	[0.31, 0.34]	<0.31	0.1430
8	Ecological benefit	The development and use rate of groundwater	X8	<1.15	[1.15, 1.19]	>1.19	0.0969
9		Maximum nitrate concentration in groundwater (mg/L)	X9	<20.89	[20.89, 33.59]	>33.59	0.1032

**Table 2 entropy-24-00920-t002:** Classification standard of surface water quality index.

Water Quality Classification	NH_3_–N (mg/L)	COD (mg/L)	Reference Standard
Class I water	≤0.15	≤15	Environmental quality standards for surface water (GB 3838-2002)
Class II water	≤0.50	≤15	
Class III water	≤1.00	≤20	
Class IV water	≤1.50	≤30	
Class V water	≤2.00	≤40	
Class V_1_ water	≤8	≤50	Discharge standard of pollutants for municipal wastewater treatment plant (GB 18918-2002)
Class V_2_ water	≤15	≤60	
Class V_3_ water	≤25	≤100	
Class V_4_ water	>25	>100	

**Table 3 entropy-24-00920-t003:** Applicability evaluation of surface water quality.

Serial Number	Types of Water Used	NH_3_–N Limits (mg/L)	COD Limits (mg/L)	NH_3_–N Concentration (mg/L)	COD Concentration (mg/L)	Suitability Evaluation Results	Reference Standard
1	Domestic water	≤0.5	≤0.5	14.46–41.80	58.00–153.39	no	Water quality standard of urban water supply (CJ/T 206-2005)
2	Industrial water	≤10	≤60			no	The reuse of urban recycling water—water quality standard for industrial uses (GB/T 19923-2005)
3	Agricultural water	/	200			yes	Standard for irrigation water quality (GB 5084-2021)
4	Economic water	5	/			no	The reuse of urban recycling water—water quality standard for scenic environmental use (GB/T 18921-2019)

**Table 4 entropy-24-00920-t004:** HV index calculation results of NSGA-II, NSGA-III, and MOEA/D algorithms.

Test Algorithm	Objective Number	Decision Variable	Mean of HV	Standard Deviation of HV
NSGA-II	2	10	0.2924	0.0192
NSGA-III	2	10	0.3024	0.0069
MOEA/D	2	10	0.2691	0.0074

**Table 5 entropy-24-00920-t005:** The actual water consumption in 2016 and results of optimal water resource allocation schemes for groundwater exploitation modes 1–3.

Groundwater Extraction Mode	Type of Water Sources	Domestic Water (10^4^ m^3^)	Industrial Water (10^4^ m^3^)	Agriculture Water (10^4^ m^3^)	Ecological Water (10^4^ m^3^)	Total Water Consumption (10^4^ m^3^)	Available Water Volume (10^4^ m^3^)
The actual water consumption	Surface water	0	0	0	0	0	1572
	Groundwater	6568.17	1949.19	12,635.27	238.1	21,390.73	18,524
	Reclaimed water	6.33	0	0	12,306.45	12,312.78	15,987
	Transferred water	2497	103	0	0	2600	2600
	Total water consumption	9071.5	2052.19	12,635.27	12,544.55	36,303.51	38,683
Groundwater extraction mode 1	Surface water	0	0	1572	0	1572	1572
	Groundwater	4655.03	288.69	9952.87	0	14,896.59	17,861.32
	Reclaimed water	1816.47	1763.5	1110.4	11,296.63	15,987	15,987
	Transferred water	2600	0	0	0	2600	2600
	Total water consumption	9071.5	2052.19	12,635.27	11,296.63	35,055.59	38,020.32
Groundwater extraction mode 2	Surface water	0	0	1572	0	1572	1572
	Groundwater	4853.59	144.13	9902.78	0	14,900.5	15,570.66
	Reclaimed water	1617.91	1908.06	1160.49	11,300.54	15,987	15,987
	Transferred water	2600	0	0	0	2600	2600
	Total water consumption	9071.5	2052.19	12,635.27	11,300.54	35,059.5	35,729.66
Groundwater extraction mode 3	Surface water	0	0	1572	0	1572	1572
	Groundwater	4692.13	131.97	9422.89	0	14,246.99	14,246.99
	Reclaimed water	1779.37	1920.22	996.08	11,291.33	15,987	15,987
	Transferred water	2600	0	0	0	2600	2600
	Total water consumption	9071.5	2052.19	11,990.97	11,291.33	34,405.99	34,405.99

## Data Availability

Data can be made available upon request. The statistical data can be found here: http://swj.beijing.gov.cn/zwgk (accessed on 10 May 2022).

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
