# Peer review of "Optimized Selection of Water Resource Allocation Schemes Based on Improved Connection Entropy in Beijing’s Southern Plain"

_entropy, 2022, doi:10.3390/e24070920_

Round 1

Reviewer 1 Report

Review of: Optimized selection of water resource allocation schemes based on improved connection entropy in Beijing’s southern plain

This paper address water allocation from different sources, including groundwater, reclaimed water, surface water, and foreign or imported water, towards different water uses in a region of Beijing.  The authors consider 3 case studies of groundwater availability, based on water quality constraints, and applied multi-objective models to determine optimal water resource allocations.  They then applied entropy measures, specifically a measure called connection entropy, to the observed and optimized water uses to choose an “optimal” optimized scheme.  

While the topic of the paper is interesting and it seems like a novel study of water resources allocation, there were several unclear aspects that necessitate moderate to major revisions before this would be acceptable for publication.  Mainly, a lot of the methodology is not clear as noted below and I think it will be difficult for readers to interpret the results.

Major comments:

I am not familiar with connection entropy, nor several of the other entropy measures that are mentioned (identity, difference, opposition entropy).  While the measure is referenced to a paper in the literature, more explanation needs to be given as to what this measure is and what it means, particularly in the context of the case study.  Terms like “connection number” (line 83) of water resources reallocation schemes were also very unclear to me.   Without more background on the derivation and meaningfulness of these measures, I cannot understand the results.  Especially since the data and programs are not provided directly (the statement is that data are “available upon request”), this is an important aspect that should be addressed.

In addition to more explanation of the entropy methods, I would recommend some illustrative example to go along with Equations 7-19, to better explain the different types of entropy, how they relate to the connection entropy, and what are “connections”.  As it is, I don’t know how the entropy-related results (presented in Figure 5) relate to “benefits”, and it isn’t clear why a low value is desirable.

My last major concern is the duration of the optimization to only focus on one year, 2016, when more years are available, and the narrow focus on “modes” that are related to groundwater extraction availabilities rather than a more comprehensive analysis over time or with different assumptions regarding the other water types.  As it is, it seems like the 3 case studies are very similar, and the only difference is that each successive case leads to lower groundwater availability.  A large part of the paper (Figures 2-4) also focuses on the water quality indexes over the basin, and it is not clear how these spatial and temporal differences have an effect on either the optimization or the entropy-based analyses (since the optimization only focuses on one year).

Minor comments:

Line 45-48: very long sentence, recommend to split

Line 86: replace “realizes” with “explores”?

Line 91: This is a bit confusing, that you optimally allocate resources according to a multi-objective model, and then optimize the optimizations using entropy?

Equations 1-3: These equations seem to be repeated twice…delete one version

Line 197: get rid of one “sigma” 

Line 220: AHP undefined? Also, Entropy weight is not clear in terms of what it means.  An illustrative example throughout these equations would help with this.

Figures 2 and 3: What is meant by the x-axis label T(a)? It seems to just be “year”

Line 453: Here you note that the volume of exploitation mode 3 is 142.47, which equates to a 76.91% extraction rate.  However, in line 417, you note that the total available groundwater is 142.47.  This seems contradictory, since shouldn’t it be 100%?  I happened to notice this and did not check other numbers, but results should be easier to interpret and verify.

Tables 4 and 5: It seems like these two tables could be combined to make a figure that compares the optimization results with the actual water uses.  As it is, it is hard to compare between numbers in the two tables.

Line 490: This sentence seems contradictory: “total benefit..decreased…indicating that it was growing” What was growing?

Figure 5: Should the labels be “groundwater exploitation mode” rather than “governance model”?  This figure is also confusing because there are 4 versions of 2016, the actual and the different model results.  I would recommend splitting this figure into 2 panels, one to show the history over time, and the other to compare the different benefits from the models.  As noted previously, the term “benefit” is also not clear from the text and how it relates to the entropy values.

Author Response

Dear Reviewer:

Thanks for your comments concerning our manuscript entitled “Optimized selection of water resource allocation schemes based on improved connection entropy in Beijing’s southern plain” (ID: 1766631). Those comments are all valuable and helpful for revising and improving our paper, as well as the important guiding significance to our researches. We have studied comments carefully and have made correction which we hope meet with approval. Revised portion are marked in the paper. The main corrections in the paper and the responds to your comments are as flowing:

Point 1: I am not familiar with connection entropy, nor several of the other entropy measures that are mentioned (identity, difference, opposition entropy).  While the measure is referenced to a paper in the literature, more explanation needs to be given as to what this measure is and what it means, particularly in the context of the case study.  Terms like “connection number” (line 83) of water resources reallocation schemes were also very unclear to me.   Without more background on the derivation and meaningfulness of these measures, I cannot understand the results.  Especially since the data and programs are not provided directly (the statement is that data are “available upon request”), this is an important aspect that should be addressed.

Response 1: Thanks for your careful review. We have added the introduction and references of connection entropy, and added the calculation results of connection number and connection entropy in the supplementary materials. Please find them below:

Line 86-98: The connection entropy was first proposed by Zhao on the basis of set pair analysis in 1992[29]. It is further derived by calculating the connection component of the connection number and is an important adjoint function of the connection number. Connection entropy is applicable to measure the order and disorder of events, reflecting the internal disorder of set pair events[26,27]. Connection entropy can reflect the connection degree of set-pair events through identify entropy, opposition entropy and difference entropy, which represent the entropy of identify degree, opposition degree and the difference degree between the evaluation samples and the evaluation standards. As a method combining set pair analysis and entropy, connection entropy can deal with system uncertainty as well as reflect the ordered development state in the system[29]. In summary, the connection entropy has unique advantages for the treatment and evaluation of water resources systems[26-29]. Therefore, the connection entropy method is selected in this paper.

line 257-272: The connection entropy is a method that combines set pair analysis and entropy. Therefore, connection entropy has the characteristics of set pair analysis to deal with uncertain problems in the system, and it can also reflect the orderly development of the system[29].

Set pair analysis, first proposed by Zhao in 1989, is commonly used to deal with the uncertain problems[29,39,40]. By establishing the connection number between samples and rules or levels to evaluate the system, which can be evaluate qualitatively or quantitatively, so that the evaluation results are more objective.

 Please find Table S1-S3 in supplementary materials.

Point 2: In addition to more explanation of the entropy methods, I would recommend some illustrative example to go along with Equations 7-19, to better explain the different types of entropy, how they relate to the connection entropy, and what are “connections”.  As it is, I don’t know how the entropy-related results (presented in Figure 5) relate to “benefits”, and it isn’t clear why a low value is desirable.

Response 2: Thanks for your comment. The connection number and connection entropy calculated according to formula 8-20 were supplemented in the supplementary materials. In this paper, the entropy values of the different water resources allocation schemes’ benefit were evaluated by connection entropy method . So it was expressed by benefit entropy value.

 Point 3: My last major concern is the duration of the optimization to only focus on one year, 2016, when more years are available, and the narrow focus on “modes” that are related to groundwater extraction availabilities rather than a more comprehensive analysis over time or with different assumptions regarding the other water types.  As it is, it seems like the 3 case studies are very similar, and the only difference is that each successive case leads to lower groundwater availability.  A large part of the paper (Figures 2-4) also focuses on the water quality indexes over the basin, and it is not clear how these spatial and temporal differences have an effect on either the optimization or the entropy-based analyses (since the optimization only focuses on one year).

Response 3: Thank for this valuable suggestion. Since the geological conditions of the study area are relatively uniform, the mining wells mainly exploit and utilize quaternary pore water, and the water consumption structure and water supply sources of the study area are relatively stable for many years without significant change. 2016 was selected as a typical year for optimization, which can reflect the basic situation of the study area.

Line 491-495: Since the geological conditions of the study area are relatively uniform, the mining wells mainly exploit and utilize quaternary pore water, and the water consumption structure and water supply sources of the study area are relatively stable for many years without significant change. 2016 was selected as a typical year for optimization, which can reflect the basic situation of the study area.

Point 4: Line 45-48: very long sentence, recommend to splitvery long sentence, recommend to split

Response 4: Thanks very much for your valuable comment. The sentence has been split. Please find it below:

Line 50-54: Although reclaimed water is increasingly important, more than 90% of reclaimed water is used for river water supplement. Because of insufficient coverage of reclaimed water pipelines or people's psychological effect, reclaimed water is not used in other aspects, resulting in waste of water resources.

Point 5: Line 86: replace “realizes” with “explores”?

Response 5: Thanks for your careful suggestions. “realizes” has been replaced with “explores”. Please find it below:

 Line 110: This study explores sustainable groundwater use by taking Beijing’s Daxing District as an example.

Point 6: Line 91: This is a bit confusing, that you optimally allocate resources according to a multi-objective model, and then optimize the optimizations using entropy?

Response 6: Thanks for your careful review. We optimally allocate resources according to a multi-objective model, and then the water resource allocation schemes were optimized selection based on the improved connection entropy.

 Line 116-117: Finally, the water resource allocation schemes were optimized selection based on the improved connection entropy.

Point 7: Equations 1-3: These equations seem to be repeated twice…delete one version

Response 7: Thanks for your suggestion. We have been deleted one version. Please find them in line 167, line 179, line 191.

Point 8: Line 197: get rid of one “sigma”

Response 8: Thanks for your suggestion. We have been deleted one “sigma”. Please find it in line 246.

Point 9: Line 220: AHP undefined? Also, Entropy weight is not clear in terms of what it means.  An illustrative example throughout these equations would help with this.

Response 9: Thanks very much for your valuable comment. The AHP is analytic hierarchy process. We have supplemented the introduction of entropy weight method and analytic hierarchy process (AHP). Please find it below:

 Line 288-303: Analytic hierarchy process (AHP) is a subjective weighting method[42], and its advantage lies in that the weight can be assigned by subjective judgment under the condition of insufficient sample data, so this method is not very ideal in terms of credibility. Entropy weight method is an objective weighting method[28], which makes full use of the original data information entropy. The reliability is ideal under the condition of high completeness of the sample data, but it has a slight disadvantage in reflecting the knowledge and experience of experts and the opinions of decision makers. Therefore, by combining the advantages and disadvantages of these two methods, this paper uses the combination weighting method of entropy weight method and analytic hierarchy process to determine the index weight.

Point 10: Figures 2 and 3: What is meant by the x-axis label T(a)? It seems to just be “year”.

 Response 10: Thanks for your valuable comment. The the x-axis label should be year in Figures 2 and 3. It has been corrected in Figures 2 and 3.

Point 11: Line 453: Here you note that the volume of exploitation mode 3 is 142.47, which equates to a 76.91% extraction rate.  However, in line 417, you note that the total available groundwater is 142.47.  This seems contradictory, since shouldn’t it be 100%?  I happened to notice this and did not check other numbers, but results should be easier to interpret and verify.

Response 11: Thanks very much for your kind suggestion. In order to compare the groundwater development rate of water resource allocation schemes easily, the rate of groundwater development refers to the ratio of the groundwater exploitation amount to the maximum available water volume which is 185.24 million m3.

 Line 596-599: In order to compare the groundwater development rate of water resource allocation schemes easily, the rate of groundwater development refers to the ratio of the groundwater exploitation amount to the maximum available water volume which is 185.24 million m3.

Point 12: Tables 4 and 5: It seems like these two tables could be combined to make a figure that compares the optimization results with the actual water uses.  As it is, it is hard to compare between numbers in the two tables.

Response 12: Thanks very much for your valuable suggestion. Tables 4 and 5 have been combined to make a table.

Point 13: Line 490: This sentence seems contradictory: “total benefit..decreased…indicating that it was growing” What was growing?

Response 13: Thanks for your valuable comment. This sentence has been corrected to “the entropy value of the actual water-use schemes’ total benefit from 2006 to 2016 decreased, from 1.40 in 2006 to 0.63 in 2016, indicating that the actual water-use schemes’ total benefitit was growing”.

Line 640-642: the entropy value of the actual water-use schemes’ total benefit from 2006 to 2016 decreased, from 1.40 in 2006 to 0.63 in 2016, indicating that the actual water-use schemes’ total benefit was growing.

Point 14: Figure 5: Should the labels be “groundwater exploitation mode” rather than “governance model”?  This figure is also confusing because there are 4 versions of 2016, the actual and the different model results.  I would recommend splitting this figure into 2 panels, one to show the history over time, and the other to compare the different benefits from the models.  As noted previously, the term “benefit” is also not clear from the text and how it relates to the entropy values.

Response 14: Thanks for your valuable comment. Figure 5 has been split into two figure. one to show the history over time, and the other to compare the different benefits from the models.

In this paper, the entropy values of the of different water resources allocation schemes’ benefit were evaluated by using the connection entropy method . So it was expressed by benefit entropy.

Once again, thank you very much for your comments and suggestions.

Yours Sincerely

Chen Li, Baohui Men, Shiyang Yin

Reviewer 2 Report

The manuscript has some merit, but it needs to be further ameliorated before being accepted. Please see my general and specific comments

General comments:

The abstract very confusing and there is no clear meaning. It should be radically reformed.

What is the added value of using an entropy optimization model instead of other approaches?

The manuscript is lacking a discussion part.

The paper does not provide comparisons (the discussion is missing) of the methodological outputs with other methodological approaches which aim at optimization problems.

Why the groundwater quality is not an objective function?

Specific comments

Lines 9-10: “Several water sources’ water supply capacity must be further excavated to improve the groundwater environment” In which way the further excavation of the water sources will improve the groundwater environment? There is no meaning in this sentence.

Lines 10-11: “achieve sustainable water resources” Of what? Sustainable water resources management? Development? The meaning is not clear.

Line 11: “the principle of quality-divided water supply” This principle is rather unknown, the authors should refer to it when they are going to further explain it within the manuscript.

Lines 13-14: “groundwater extraction modes”. Is not clear what a groundwater extraction mode is?

Lines 23-24: “use rate of groundwater development” What is the use rate of groundwater development?

Lines 27-28: “and actual water-use schemes from 2006 to 2016.” The sentence cannot be understood. The research has to do with sampling from 2012 to 2016. Why the authors refer to 2006?

Line 43: “According to the Beijing Water Statistics Yearbook”. Of what year?

Line 52: “the principle of quality-divided water supply” This principle should be explained in more details since it is not broadly known. Scientific references of the specific principle should be provided.

Line 61. “many scholars will consider more details”. I propose to omit the “will”

Line 63: I would propose the following sentence to be added at the end of the specific sentence. “Other scholars propose fuzzy AHP-outranking frameworks for the prioritization of measures focused on the protection of the water resources including water supply sources (Spiliotis and Skoulikaris, 2019)”

where Spiliotis, M., & Skoulikaris, C. (2019). A fuzzy AHP-outranking framework for selecting measures of river basin management plans. Desalination Water Treat167, 398-411.

Line 116. Statistical data. Are the data annual averages or monthly mean values. The authors should state the frequency sample.

Line 116: 2.2.2 Statistical data. References in form of URLs should be at least provided for the validation of the utilized data sources.

Line 132. The authors use 2 objective functions related to the “minimum total water shortage” and the “minimum water cost”. Why the water quality is not an objective function (or rather two objective functions for surface water and groundwater) , since the quality of the water is the main problem within the case study area?

Line 145: “foreign water” What is foreign water?

Line 174: “Water consumption constraint of using water departments”. What are the water departments? There is not such an expression.

Lines 190-192: The authors should provide a synoptically description for each of the three utilized algorithms, in order to better understand the differences among them.  

Line 197: “σ?”???

Lines 267-268: “and ten-thousand-yuan industrial output water consumption.” It cannot be understood.

Line 219: “The ecological benefits include the usage rate of groundwater development”. It cannot be understood what the groundwater development is. What the authors do mean? They must be more precise.

Line 228: “and expert opinions”. Who are these experts?

Line 231. “levels I, II, and III”?? Which is better?

Lines 404-418: The authors present some figures regarding the water volume that is related with thw concentrations of NO3-N. For example in mode 1, the authors say “The amount of water available for groundwater decreased by 6.63 million m3, and the available water for groundwater was 178.61 million m3.” Where did these figures come from? Do the authors know all the wells and the derived pumping volumes that are used for groundwater extraction and use? More information on the way that these figures are coming from are required.

Author Response

Dear Reviewer:

Thanks for your comments concerning our manuscript entitled “Optimized selection of water resource allocation schemes based on improved connection entropy in Beijing’s southern plain” (ID: 1766631). Those comments are all valuable and helpful for revising and improving our paper, as well as the important guiding significance to our researches. We have studied comments carefully and have made correction which we hope meet with approval. Revised portion are marked in the paper. The main corrections in the paper and the responds to your comments are as flowing:

General comments:

Point 1: The abstract very confusing and there is no clear meaning. It should be radically reformed.

 Response 1: Thanks very much for your valuable comment. The abstract has been radically reformed.

Line 8-34: The increased urbanization has caused problems, such as increasing water consumption and continuous deterioration of the groundwater environment. It is necessary to consider the groundwater quality in the water resource optimization system, and increase the rate of reclaimed water development to reduce the amount of groundwater exploitation and achieve sustainable water resources development. This study used the Daxing District, a region of Beijing’s southern plain, as an example to evaluate the water quality by collecting water quality data of surface and groundwater from 2012 to 2016, and actual water-use schemes from 2006 to 2016. Three groundwater extraction modes were set up based on NO3–N concentrations, water resources were optimized under three extraction modes, and water resource optimization schemes were determined based on the improved connection entropy. The results show that (1) the surface water quality was poor, and the proportion of V4 type water in the indexes of NH3–N and chemical oxygen demand (COD) was the largest. The surface water can only be used for agricultural irrigation. The pollution sources contributing most to NH3–N and COD were domestic and agricultural pollution sources. (2) The groundwater quality was good. The NO3–N index was primarily type I–III water, accounting for 95.20% of the total samples. Severe NH3–N pollution areas were mainly in the northern region, and most regional groundwater can be used for various purposes. (3) Taking 2016 as an example, three groundwater exploitation modes were set to optimize water resource allocation, and the results showed that the rate of groundwater development and NO3–N pollution decreased significantly after optimization. (4) Based on the improved connection entropy, the connection entropy of water resource optimal allocation and actual water-use schemes were calculated. The connection entropy of groundwater exploitation mode 3 was less than that of groundwater exploitation modes 1 and 2 and actual water-use schemes from 2006 to 2016. Therefore, exploitation mode 3’s water resource optimization scheme was recommended. This study’s research methods and outcomes can provide methodological and theoretical lessons for water management in freshwater deficient areas.

Point 2: What is the added value of using an entropy optimization model instead of other approaches?

Response 2: Thanks for your valuable comment. The value of using an entropy optimization model has been added.

 Line 86-98: The connection entropy was first proposed by Zhao on the basis of set pair analysis in 1992[29]. It is further derived by calculating the connection component of the connection number and is an important adjoint function of the connection number. Connection entropy is applicable to measure the order and disorder of events, reflecting the internal disorder of set pair events[26,27]. Connection entropy can reflect the connection degree of set-pair events through identify entropy, opposition entropy and difference entropy, which represent the entropy of identify degree, opposition degree and the difference degree between the evaluation samples and the evaluation standards. As a method combining set pair analysis and entropy, connection entropy can deal with system uncertainty as well as reflect the ordered development state in the system[29]. In summary, the connection entropy has unique advantages for the treatment and evaluation of water resources systems[26-29]. Therefore, the connection entropy method is selected in this paper. Because connection entropy cannot clearly reflect the physical connotation of identity, difference, and opposition entropy, this paper adopts a method for improving the connection entropy. We first calculate the samples’ connection numbers and membership degrees, determine the difference coefficient using the proportional value method, and finally obtain the total entropy value of the connection number of each water resource allocation scheme. The optimal water resource allocation scheme can be obtained by comparing the total entropy of the connection number.

Point 3: The manuscript is lacking a discussion part.

Response 3: Thanks very much for your valuable suggestion. The discussion about three optimization algorithms has been added.

Line 560-567: Table 3 shows that NSGA-III was the largest mean value and the smallest standard deviation of HV, indicating that NSGA-III was better performance in solving the established multi-objective optimization model compared with NSGA-II and MOEA/D. Compared with MOEA/D, the HV mean of NSGA-II was larger than that of MOEA/D, but the HV standard deviation of NSGA-II was larger than that of MOEA/D, indicating that the performance difference between NSGA-II and MOEA/D was small, but the performance of NSGA-II and MOEA/D were inferior to that of NSGA-III. Therefore, we chose NSGA-III as the solution algorithm of the optimization model in this paper.

Point 4: The paper does not provide comparisons (the discussion is missing) of the methodological outputs with other methodological approaches which aim at optimization problems.

Response 4: Thanks very much for your kind suggestion. The discussion about three optimization algorithms has been added.

Line 560-567: Table 3 shows that NSGA-III was the largest mean value and the smallest standard deviation of HV, indicating that NSGA-III was better performance in solving the established multi-objective optimization model compared with NSGA-II and MOEA/D. Compared with MOEA/D, the HV mean of NSGA-II was larger than that of MOEA/D, but the HV standard deviation of NSGA-II was larger than that of MOEA/D, indicating that the performance difference between NSGA-II and MOEA/D was small, but the performance of NSGA-II and MOEA/D were inferior to that of NSGA-III. Therefore, we chose NSGA-III as the solution algorithm of the optimization model in this paper.

Point 5: Why the groundwater quality is not an objective function?

Response 5: Thanks for your valuable comment. In this paper, two objective functions were set up—the minimum total water shortage and the minimum total water cost. Although the groundwater quality is not used as the objective function, according to the groundwater quality, the amount of groundwater extraction in areas with poor groundwater quality is limited. That is to say, groundwater quality has become a constraint condition of groundwater supply capacity. Three groundwater exploitation modes are set up in Section 3.2, that is, water resource allocation schemes are carried out under the three groundwater exploitation modes.

Specific comments:

Point 6: Lines 9-10: “Several water sources’ water supply capacity must be further excavated to improve the groundwater environment” In which way the further excavation of the water sources will improve the groundwater environment? There is no meaning in this sentence.

Response 6: Thanks very much for your valuable comment. This sentence has been corrected.

Line 9-12: It is necessary to consider the groundwater quality in the water resource optimization system, and increase the rate of reclaimed water development to reduce the amount of groundwater exploitation and achieve sustainable water resources development.

 Point 7: Lines 10-11: “achieve sustainable water resources” Of what? Sustainable water resources management? Development? The meaning is not clear.

Response 7: Thanks for your suggestion. It should be “achieve sustainable water resources development ”. Please find it in line 11-12.

Point 8: Line 11: “the principle of quality-divided water supply” This principle is rather unknown, the authors should refer to it when they are going to further explain it within the manuscript.

Response 8: Thanks for your careful review. “the principle of quality-divided water supply” has been corrected to “water supply with different quality”. And we have added the references. Please find “water supply with different quality” in line 61.

Point 9: Lines 13-14: “groundwater extraction modes”. Is not clear what a groundwater extraction mode is?

Response 9: Thanks for your careful suggestions. This sentence is corrected as follows:

Line 17-19: Three groundwater extraction modes were set up based on NO3–N concentrations, water resources were optimized under three extraction modes, and water resource optimization schemes were determined based on the improved connection entropy.

Point 10: Lines 23-24: “use rate of groundwater development” What is the use rate of groundwater development?

Response 10: Thanks for your careful review. “use rate of groundwater development” has been corrected to “the rate of groundwater development”. Please find it in line 27.

Point 11: Lines 27-28: “and actual water-use schemes from 2006 to 2016.” The sentence cannot be understood. The research has to do with sampling from 2012 to 2016. Why the authors refer to 2006?

Response 11: Thanks for your careful suggestions. By collecting the water quality data of surface water and groundwater in the study area from 2012 to 2016, its applicability was evaluated and analyzed. The statistical and actual water consumption data from 2006 to 2016 were collected to select the water resource allocation schemes. The water resource allocation schemes for three groundwater exploitation modes were compared with the actual water consumption from 2006 to 2016, and the best scheme was selected.

Line 14-19: This study used the Daxing District, a region of Beijing’s southern plain, as an example to evaluate the water quality by collecting water quality data of surface and groundwater from 2012 to 2016, and actual water-use schemes from 2006 to 2016. Three groundwater extraction modes were set up based on NO3–N concentrations, water resources were optimized under three extraction modes, and water resource optimization schemes were determined based on the improved connection entropy.

Point 12: Line 43: “According to the Beijing Water Statistics Yearbook”. Of what year?

Response 12: Thanks very much for your valuable comment. The time has been added. This sentence is corrected as follows:

Line 48-50: According to the Beijing Water Statistics Yearbook from 2003 to 2020, the proportion of reclaimed water supply in Beijing increased from 5.73% to 29.56%, indicating that reclaimed water is vital in the existing water supply pattern.

Point 13: Line 52: “the principle of quality-divided water supply” This principle should be explained in more details since it is not broadly known. Scientific references of the specific principle should be provided.

Response 13: Thank for this valuable suggestion. In order to express more accurately, “the principle of quality-divided water supply”has been corrected to “water supply with different quality”, and relevant references have been added according to expert opinions. Please find it in line 61.

Point 14: Line 61. “many scholars will consider more details”. I propose to omit the “will”

Response 14: Thanks for your careful review. This sentence is corrected as follows:

Line 71: With further research on optimal water resource allocation, many scholars consider more details, such as the fairness of water supply[16], the risk of water supply shortage[17,18], and other factors.

Point 15: Line 63: I would propose the following sentence to be added at the end of the specific sentence. “Other scholars propose fuzzy AHP-outranking frameworks for the prioritization of measures focused on the protection of the water resources including water supply sources (Spiliotis and Skoulikaris, 2019)”

where Spiliotis, M., & Skoulikaris, C. (2019). A fuzzy AHP-outranking framework for selecting measures of river basin management plans. Desalination Water Treat, 167, 398-411.

Response 15: Thanks for your careful review and kind suggestion. It has been added. Please find it in line 72-74.

Line 72-74: Other scholars propose fuzzy AHP-outranking frameworks for the prioritization of measures focused on the protection of the water resources including water supply source[19].

Point 16: Line 116. Statistical data. Are the data annual averages or monthly mean values. The authors should state the frequency sample.

Response 16: Thanks for your careful review. The statistical data is annual data. It has been corrected in the revised manuscript. Please find it in line 141-142.

Point 17: Line 116: 2.2.2 Statistical data. References in form of URLs should be at least provided for the validation of the utilized data sources.

Response 17: Thanks for your careful and kind review. References in form of URLs have be provided for the validation of the utilized data sources. It can be found in line 716-717.

Line 716-717: Data Availability Statement: Data can be made available upon request. The statistical data can be found here: http://swj.beijing.gov.cn/zwgk.

Point 18: Line 132. The authors use 2 objective functions related to the “minimum total water shortage” and the “minimum water cost”. Why the water quality is not an objective function (or rather two objective functions for surface water and groundwater) , since the quality of the water is the main problem within the case study area?

Response 18: Thanks for your kind comment. In this paper, two objective functions were set up—the minimum total water shortage and the minimum total water cost. Although the groundwater quality is not used as the objective function, according to the groundwater quality, the amount of groundwater extraction in areas with poor groundwater quality is limited. That is to say, groundwater quality has become a constraint condition of groundwater supply capacity. Three groundwater exploitation modes are set up in Section 3.2, that is, water resource allocation schemes are carried out under the three groundwater exploitation modes.

Point 19: Line 145: “foreign water” What is foreign water?

Response 19: Thanks for your valuable suggestion. The foreign water has been corrected to “transferred water”. It means water transferred from other places, not local water. Please find it in line 171.

Point 20: Line 174: “Water consumption constraint of using water departments”. What are the water departments? There is not such an expression.

Response 20: Thanks for your comment. “Water consumption constraint of using water departments” has been corrected to “Water consumption constraint”. Please find it in line 202.

Point 21: Lines 190-192: The authors should provide a synoptically description for each of the three utilized algorithms, in order to better understand the differences among them. 

Response 21: Thanks for your careful review. We have supplemented the synoptically description for each of the three utilized algorithms, in order to better understand the differences among them. It can be found as follows:

Line 221-241: Deb proposed the NSGA-II algorithm in 2002, mainly to improve the shortcomings of the NSGA algorithm[34]. Compared with NSGA, NSGA-II adopts fast non-dominated sorting method, which greatly reduces the computation time. The elite strategy ensures that excellent individuals can be retained with greater probability. Crowding method was used instead of fitness sharing strategy, which needed to ensure the diversity of individual. The algorithm frameworks of NSGA-III and NSGA-II are roughly same, but the selection mechanism is different[35]. NSGA-II uses crowding to select individuals with the same non-dominated level. While NSGA-III selects individuals based on reference points. NSGA-III adopts the method based on reference points to solve the problem of poor convergence and diversity of the algorithm if the crowding distance is continued in the face of multi-objective optimization problems with three or more objectives.

In order to find the pareto optimal solution of the multi-objective optimization problem, Zhang proposed MOEA/D[36]. The core idea of this method is to transform the multi-objective optimization problem into a series of single-objective optimization sub-problems or multiple multi-objective sub-problems, and then use the neighborhood relationship between sub-problems to optimize these sub-problems in a collaborative manner, so as to approach the entire pareto front. Usually, the definition of sub-problems is determined by the weight vector, and the neighborhood relationship between subproblems is determined by calculating the euclidean distance between the weight vectors.

Point 22: Line 197: “σ?”??? 

Response 22: Thanks for your careful review. We have deleted one of σ. Please find it in line 246.

Point 23: Lines 267-268: “and ten-thousand-yuan industrial output water consumption.” It cannot be understood. 

Response 23: Thanks for your comment.” ten-thousand-yuan industrial output water consumption” has been corrected to “water consumption of ten thousand yuan industrial output value”. Please find it in line 282-283.

Point 24: Line 219: “The ecological benefits include the usage rate of groundwater development”. It cannot be understood what the groundwater development is. What the authors do mean? They must be more precise.

Response 24: Thanks for your comment. “the usage rate of groundwater development” has been corrected to “the rate of groundwater development”. Please find it in line 286.

Point 25: Line 228: “and expert opinions”. Who are these experts?

Response 25: Thanks for your careful review. We have removed expert opinions and added related literatures. Please find it in line 308-309.

Point 26: Line 231. “levels I, II, and III”?? Which is better?

Response 26: Thanks for your kind comment. Levels I is better. Level I is good, level II is medium, level III is poor. It can be found in the first row of Table 5.

Point 27: Lines 404-418: The authors present some figures regarding the water volume that is related with the concentrations of NO3-N. For example in mode 1, the authors say “The amount of water available for groundwater decreased by 6.63 million m3, and the available water for groundwater was 178.61 million m3.” Where did these figures come from? Do the authors know all the wells and the derived pumping volumes that are used for groundwater extraction and use? More information on the way that these figures are coming from are required.

Response 27: Thanks for your careful and kind review. The amount of water available for groundwater in mode 1-3 has been supplemented in the revised manuscript. Please find it below:

 Line 498-509: Groundwater exploitation mode 1: The areas where groundwater NO3–N concentrations were above 20 mg/L were not regarded as domestic water.

As can be seen from Figure 4 (b), the areas with NO3-N concentrations above 20mg/L were mainly located in Jiugong, Yinghai and Yizhuang, covering an area of 35.67km2. The area with NO3-N above 20mg/L account for 59.92%, 30.35% and 19.34% of the total area of Jiugong, Yinghai and Yizhuang, respectively. According to the actual domestic water consumption in 2016, and the above area ratio, it can be calculated that the groundwater exploitation volumes of the area with NO3-N concentrations above 20mg/L in Jiugong, Yinghai and Yizhuang were 5.00 million m3, 1.10 million m3 and 0.53 million m3 respectively, totaling 6.63 million m3. Therefore, according to groundwater exploitation mode 1, the available groundwater supply decreased by 6,63 million m3, and the available groundwater supply was 178.61 million m3.

Line 512-531: Groundwater exploitation mode 2: Groundwater in areas with NO3–N concentrations of 10 mg/L–20 mg/L was not used as domestic water, and groundwater exploitation in areas with NO3–N concentrations above 20 mg/L was prohibited.

Figure 4 (b) shows that the areas with NO3-N concentrations between 10mg/L and 20mg/L were mainly located in Huangcun, Yinghai, Xihongmen, Jiugong and Yizhuang, covering an area of 166.77 km2. And the area of NO3-N concentrations in 10 mg/L ~ 20 mg/L accounted for 59.81%, 30.69%, 100%, 40.08% and 80.66% of the above each township’s total area, respectively. According to the actual domestic water consumption in Huangcun, Yinghai, Xihongmen, Jiugong and Yizhuang in 2016, those areas’ groundwater exploitation of NO3-N concentrations in 10 mg/L ~ 20 mg/L were 9.09 million m3, 1.11 million m3, 5.52 million m3, 3.34 millon m3, 2.23 million m3, a total of 21.29 million m3.

From the above, the area of NO3-N concentrations above 20 mg/L accounted for 59.92 %, 30.35 %, 19.34 % of the total area of Jiugong, Yinghai, Yizhuang. According to the actual groundwater exploitation of Jiugong, Yinghai and Yizhuang in 2016, the groundwater exploitation of those areas with NO3-N above 20 mg/L were 5.78 million m3, 1.75 million m3, 0.71 million m3, a total of 8.24 million m3 by the above area ratio.

Therefore, according to groundwater exploitation mode 2, groundwater available water supply decreased by 29.53 million m3 and groundwater available water supply is 155.71 million m3.

Line 534-546: Groundwater exploitation mode 3: Groundwater exploitation was prohibited in areas where NO3–N concentrations were above 10 mg/L.

Figure4 (b) shows that the areas of NO3-N above 10mg/L mainly located in Huangcun, Yinghai, Xihongmen, Jiugong, Yizhuang, and the total area was 202.44km2. The area of NO3-N above 10 mg/L accounted for 59.81 %, 61.04 %, 100 %, 100 % and 100 % of Huangcun, Yinghai, Xihongmen, Jiugong and Yizhuang, respectively. According to the actual groundwater exploitation of Huangcun, Yinghai, Xihongmen, Jiugong and Yizhuang in 2016 and the above area ratio, the groundwater exploitation of Huangcun, Yinghai, Xihongmen, Jiugong and Yizhuang with NO3-N concentrations above 10 mg/L were 17.98 million m3, 3.52 million m3, 7.95 million m3, 9.64 million m3, 3.68 million m3, a total of 42.77 million m3. Therefore, according to groundwater exploitation mode 3, groundwater available water supply decreased by 42.77 million m3 and groundwater available water supply was 142.47 million m3.

Once again, thank you very much for your comments and suggestions.

Yours Sincerely

Chen Li, Baohui Men, Shiyang Yin

Round 2

Reviewer 1 Report

The authors have made several efforts to improve this manuscript and have addressed most of my previous comments.  With this, I have several items that constitute minor revisions that would make this a good contribution to the journal Entropy.  While I appreciate the effort by the authors to better explain the methods, there is still quite a bit of jargon involved and I will try to make some specific suggestions to make it more accessible.

Main comment: 

Some of the methods, particularly the connection entropy, is still not intuitive to understand without reading many other papers on the topic.  For example, below is a list of terms or phrases that are introduced but not fully explained to the reader, or explained much later in the paper:

Line 29: connection entropy (has not been at all defined in the abstract)

Line 73: AHP (the acronym is now defined, but much later)

Line 80: "standards of identify, difference, and opposition".  I think these terms are abstract and impossible to appreciate without context.  For example, difference between what and what? Or opposition to what?

Line 81: What is the "evaluation dimension"? and how can it be "abundant"?

Line 88: "connection component of the connection number" is not clear at this point.  What is the connection number, and what is the connection component?  Some illustration (based on the types of variables you are considering) or extra sentences for connotation here would go a long way to help with this interpretation.

Line 106: membership degree, difference coefficient, and proportional value method are all jargon terms that a reader is not likely to understand without context

Line 262: "the uncertain problems" is unclear - what types of uncertainty-related problems?

Line 275: In general, the Steps (Step 1, Step 2, Step 3 sections) are rather abstract and it is difficult to connect them to the actual data set being used.

Line 319: "sample values" - what is a sample? is it a data point of an evaluation metric, or something else?

Line 703: This is a useful paragraph and I think it would solve several problems to have it earlier in the paper when those terms are first introduced.  Why wait until the very end to give any interpretation of what is meant by these different types of entropy?

Finally, it is good that the authors added some supplemental tables.  However, these are not described at all in the SI or the main text, so at least a paragraph should be added in either place to more directly describe what these tables are providing and how they help the interpretation of the main figures.

Minor comments:

Line 15: "by collecting" makes it sound like part of this study was the actual data collection.  Suggest to change to "by analyzing"

Line 66: not sure what "inconducive" means here..."not a priority"?

Line 263: evaluate --> evaluated

Line 271: i should be upper case I to match equation

Line 301: jth index, J should be lower case to match equation

Author Response

Dear Reviewer:

Thanks for your comments concerning our manuscript entitled “Optimized selection of water resource allocation schemes based on improved connection entropy in Beijing’s southern plain” (ID: 1766631). Those comments are all valuable and helpful for revising and improving our paper, as well as the important guiding significance to our researches. We have studied comments carefully and have made correction which we hope meet with approval. Revised portion are marked in the paper. The main corrections in the paper and the responds to your comments are as flowing:

Point 1: Line 29: connection entropy (has not been at all defined in the abstract)

Response 1:  Thanks for your careful review. We have added the connection entropy in the abstract. Please find it below:

 Line 28-38: Connection entropy is an evaluation method that combines connection numbers and entropy, including identify, difference, and opposition entropy. Due to the connection entropy is a kind of complete entropy, which can reflect the difference of the system in different states. Therefore, based on the improved connection entropy, the connection entropy of water resource optimal allocation and actual water-use schemes were calculated. The connection entropy of groundwater exploitation mode 3 was less than that of groundwater exploitation modes 1 and 2 and actual water-use schemes from 2006 to 2016. Therefore, exploitation mode 3’s water resource optimization scheme was recommended. In the paper, the satisfactory results have been obtained. As a kind of complete entropy, connection entropy has great research value in dealing with complex hydrological problems.

 Point 2: Line 73: AHP (The acronym is now defined, but much later)

Response 2:  Thanks for your suggestion. We have defined AHP below:

 Line 77-79: Other scholars propose fuzzy (Analytic hierarchy process) AHP-outranking frameworks for the prioritization of measures focused on the protection of the water resources including water supply source[19].

 Point 3: Line 80: “standards of identify, difference, and opposition”. I think these terms are abstract and impossible to appreciate without context. For example, difference between what and what? or opposition to what?

Response 3:  Thanks very much for your valuable comment. We have corrected the sentence below:

Line 84-86: Compared with previous evaluation methods, set pair analysis can reflect the identity, difference and opposition between evaluation indexes and standards.

Point 4: Line 81: What is the “evaluation dimension”? and how can it be “abundant”?

Response 4:  Thanks for your suggestion. We have supplemented the introduction of the identity, difference,and opposition of the set pair analysis. Please find it below.

 Line 87-95: A pair formed by two sets that are connected in a certain way is called set pair. If the two sets have some of the same characteristics, it is called the identity connection. If two sets have some opposite characteristics, it is called the opposition connection. If the connection between the two sets is neither an identity connection nor an opposite connection, then it is called the difference connection. Among them, identity connection and opposition connection are called the deterministic connection of two sets, and the difference connection is called the uncertainty connection. The connection degree of the identity connection, difference connection, and opposition connection between the two sets can be expressed by connection degree. Therefore, the evaluation dimension is more abundant and can display the influence of evaluation indexes on samples more comprehensively.

Point 5: Line 88: “connection component of the connection number” is not clear at this point. What is the connection number, and what is the connection component? Some illustration (based on the types of variables you are considering) or sentences for connotation here would go a long way to help with this interpretation.

Response 5:  Thanks for your valuable comment. We have added the introductions of connection number and connection component. Please find it below:

Line 296-307: The general expression of connection number is as follows (7) :

u=a+bI+cJ=u1+u2+u3       (7)

In the formula, u denotes the connection degree between two set pair events, also known as the ternary connection number. a, b and c are the degree of identity, difference and opposition of two set pair events, respectively, and the ranges of a, b and c are [0,1], and a+b+c=1 ; I represents the difference coefficient, and the value interval is [−1,1]; J denotes the opposition coefficient, J = -1. u1, u2, and u3 represent the identity degree component, difference degree component, and opposition degree component of the connection degree u, respectively.

The connection degree of the identity connection, difference connection, and opposition connection between the two sets can be expressed by connection degree u. The expression a+bI+cJ of connection degree is the connection number of the sample.

Point 6: Line 106: membership degree, difference coefficient, and proportional value method are all jargon terms that a reader is not likely to understand without context.

Response 6:  Thanks very much for your kind suggestion. The sentence has been corrected. Please find it below:

Line 128-131: We first calculate the samples’ connection numbers, then determine the difference coefficient, calculate the connection degree value of each indicator, and finally obtain the total entropy value of the connection number of each water resource allocation scheme.

Point 7: Line 262: “the uncertain problems” is unclear – what types of uncertainly-related problems?

Response 7:  Thanks for your valuable comment. The uncertain problems have been added. Please find it below:

 Line 285-292: In the current description of uncertain systems, one is the probability and statistics theory to describe random uncertainty, and the other is the fuzzy mathematical theory to describe fuzzy uncertainty. The theory of probability and statistics overemphasizes the independence of the system, while the theory of fuzzy mathematics relies too much on subjective experience, so both theories have shortcomings. Set pair analysis, first proposed by Zhao in 1989, is commonly used to deal with random uncertainty and fuzzy uncertainty problems[29,39,40]. When dealing with uncertainty, it is more objective and the operation is simpler.

 Point 8: Line 275: In general, the steps (Step 1, Step 2, Step 3 sections) are rather abstract and it is difficult to connect them to the actual data set being used.

Response 8:  Thanks for your valuable comment. We move the original table 5 to table 1 to make steps 1-3 easier to understand. Please find it below:

 Line 341-345:

The evaluation indicators and indicator weights of the opitimal water resource allocation scheme see the Table 1.

Table 1. Evaluation indicators, standard grades, and indicator weights of the optimal water resource allocation scheme

Serial number

Target

Evaluation indicators

Symbol

Indicator level

weight

Good (level I)

Medium (level II)

Poor (level III)

1

Social benefit

Water quota per capita (L/(person∙day))

X1

>154.83

[146.71, 154.83]

<146.71

0.1174

2

Agricultural water-use ratio

X2

<0.36

[0.36, 0.72]

>0.72

0.0981

3

Water consumption per ten thousand yuan of industrial output value (m3/104 yuan)

X3

<2.78

[2.78, 4.15]

>4.15

0.1010

4

Economic benefit

Total cost of water supply (100 million yuan)

X4

<13.66

[13.66, 14.42]

>14.42

0.1036

5

Groundwater supply cost (100 million yuan)

X5

<10.69

[10.69, 11.05]

>11.05

0.0970

6

Reclaimed water supply cost (100 million yuan)

X6

>1.24

[1.14, 1.24]

<1.14

0.1398

7

Reclaimed water supply ratio

X7

>0.34

[0.31, 0.34]

<0.31

0.1430

8

Ecological benefit

The development and use rate of groundwater

X8

<1.15

[1.15, 1.19]

>1.19

0.0969

9

Maximum nitrate concentration in groundwater (mg/L)

X9

<20.89

[20.89, 33.59]

>33.59

0.1032

Line 354: The level limit for each evaluation indicator sees the Table 1.

Point 9: Line 319: “sample values” – what is a sample? is a data pint of an evalution metric, or something else?

Response 9:  Thanks very much for your valuable suggestion. The sentence has been corrected. Please find it below.

 Line 360-362: Calculate the index connection number between the jth index value of the evaluation sample i and the evaluation standard of optimal water resource allocation. If the indicator is positive with xij ≤ s1j, or negative with xij ≥ s1j,.

Point 10: Line 703: this is a useful paragraph and I think it would solve several problems to have it earlier in the paper when those terms are first introduced. Why wait until the very end to give any interpretation of what it meant by these different types of entropy?

Response 10: Thanks for your valuable comment. The paragraph has been moved to Line 100-133. Please find it below:

Line 100-133: The connection entropy was first proposed by Zhao on the basis of set pair analysis in 1992[29]. It is further derived by calculating the connection component of the connection number and is an important adjoint function of the connection number. Connection entropy is applicable to measure the order and disorder of events, reflecting the internal disorder of set pair events[26,27]. Connection entropy can reflect the connection degree of set-pair events through identify entropy, opposition entropy and difference entropy, which represent the entropy of identify degree, opposition degree and the difference degree between the evaluation samples and the evaluation standards. As a method combining set pair analysis and entropy, connection entropy can deal with system uncertainty as well as reflect the ordered development state in the system[29]. The identity entropy is equal to the information entropy, which can be used to measure the uncertainty degree of the system; the opposition can be used to measure the certainty degree in the system; the difference entropy can be used to measure the part of the system where certainty and uncertainty coexist. It can also be said that the identity entropy is equal to the measurement of the disordered part, the opposition is equal to the measurement of the ordered part, and the different entropy is equal to the measurement of the intermediate state between the ordered and the disordered. Therefore, the connection entropy is a kind of complete entropy, which can reflect the difference of the system in different states. In summary, the connection entropy has unique advantages for the treatment and evaluation of water resources systems[26-29]. Therefore, the connection entropy method is selected in this paper. Because connection entropy cannot clearly reflect the physical connotation of identity, difference, and opposition entropy, this paper adopts a method for improving the connection entropy. We first calculate the samples’ connection numbers, then determine the difference coefficient, calculate the connection value of each indicator, and finally obtain the total entropy value of the connection number of each water resource allocation scheme. The optimal water resource allocation scheme can be obtained by comparing the total entropy of the connection number.

Point 11: Finally, it is good that the authors added some supplemental tables. However, these are not described at all in the S1 or the main text, so at least a paragraph should be added in either place to more directly describe what these tables are providing and how they help the interpretation of the main figures.

Response 11: Thanks very much for your valuable comment. The introductions of supplemental tables have been added. Please find it below:

Line 671-678: According to Formulas (8)–(20), the total entropy of the connection number of the actual water-use schemes from 2006 to 2016 and the optimal water resource allocation schemes for the three groundwater exploitation modes were calculated(see Table S1-S3). The table S1-S2 show the connection number of indicators of the actual water-use schemes from 2006 to 2016 and the optimal water resources allocation schemes for the three groundwater exploitation modes. The table S3 shows the connection entropy of the actual water-use schemes from 2006 to 2016 and the optimal water resources allocation schemes for the three groundwater exploitation modes.

Minor comments

Point 12: Line 15: “by collecting” makes it sound like part of this study was the actual data collection. Suggest to change to “by analyzing”.

Response 12:  Thanks for your suggestion. It has been corrected. Please find it below:

Line 14-17: This study used the Daxing District, a region of Beijing’s southern plain, as an example to evaluate the water quality by analyzing water quality data of surface and groundwater from 2012 to 2016, and actual water-use schemes from 2006 to 2016.

Point 13: Line 66: not sure what “inconducive” means here…”not a priority”?

Response 13: Thanks for your suggestion. It has been corrected. Please find it below:

Line 70-71: Due to this single consideration, it is harmful to improve water resource use efficiency[13].

Point 14: Line 263: evaluate --> evaluated

Response 14:  Thanks for your suggestion. It has been corrected. Please find it below:

Line 292-294:  By establishing the connection number between samples and rules or levels to evaluate the system, which can be evaluated qualitatively or quantitatively, so that the evaluation results are more objective.

Point 15: Line 271: i should be upper case I to match equation

Response 15: Thanks for your suggestion. It has been corrected. Please find it below:

Line 301:  I represents the difference coefficient, and the value interval is [−1,1]

Point 16: Line 301: jth index, J should be lower case to match equation

Response 16:  hanks for your suggestion. It has been corrected. Please find it below:

Line 366-338:  In the formula,  is the combined weight value of the jth index; and are the weight values of entropy weight method and analytic hierarchy process, respectively. β is the preference coefficient, taking 0.5.

Once again, thank you very much for your comments and suggestions.

Yours Sincerely

Chen Li, Baohui Men, Shiyang Yin

Reviewer 2 Report

The authors of the manuscript "Optimized selection of water resource allocation schemes based on improved connection entropy in Beijing’s southern plain" have peer reviewed the manuscript and have clearly answered to all my initial comments. Thus, I propose the acceptance of the research to be further processed for publication. 

Author Response

Dear Reviewer:

Thanks for your comments concerning our manuscript entitled “Optimized selection of water resource allocation schemes based on improved connection entropy in Beijing’s southern plain” (ID: 1766631). Those comments are all valuable and helpful for revising and improving our paper, as well as the important guiding significance to our researches.

Once again, thank you very much for your comments and suggestions.

Yours Sincerely

Chen Li, Baohui Men, Shiyang Yin